# Effects of strongly eddying oceans on multidecadal climate variability in the Community Earth System Model

André Jüling[1], Anna von der Heydt[1], and Henk A. Dijkstra[1]

[1]Institute for Marine and Atmospheric research Utrecht (IMAU), Utrecht University, Netherlands

**Correspondence:** André Jüling (a.juling@uu.nl)

**Abstract.** Climate variability on multidecadal time scales appears to be organized in pronounced patterns with clear expressions in sea surface temperature, such as the Atlantic Multidecadal Variability and the Pacific Decadal Oscillation. These patterns are now well studied both in observations and global climate models and are important in the attribution of climate change. Results from CMIP5 models have indicated large biases in these patterns with consequences for ocean heat storage variability and the global mean surface temperature. In this paper, we use two multi-century Community Earth System Model simulations at coarse (1°) and fine (0.1°) ocean model horizontal grid spacing to study the effects of the representation of mesoscale ocean flows on major patterns of multidecadal variability. We find that resolving mesoscale ocean flows both improves the characteristics of the modes of variability with respect to observations and increases the amplitude of the heat content variability in the individual ocean basins. In the strongly eddying model, multidecadal variability increases compared to sub-decadal variability. This shift of spectral power is seen in sea surface temperature indices, basin-scale surface heat fluxes, as well as the global mean surface temperature. This implies that the current CMIP6 model generation which predominantly does not resolve the ocean mesoscale may systematically underestimate multidecadal variability.

**Keywords.** Multidecadal Climate Variability

Mesoscale ocean flows

Ocean Heat Content

Global Mean Surface Temperature

## 1 Introduction

The ocean plays a key role in the climate system's heat budget, absorbing some 93% of the additional heat retained due to anthropogenic greenhouse gases (Stocker et al., 2013). Although the instrumental record of sea surface temperature (SST) is only about one and a half centuries, the observations indicate the existence of spatially correlated patterns of variability on multidecadal time scales, also referred to as (statistical) modes of variability (Deser et al., 2010). These modes are thought to be part of the internal variability of the climate system and they affect the oceanic heat content by altering heat fluxes and

consequently the global energy budget (Trenberth and Shea, 2006; Dijkstra, 2013; Zhang and Wang, 2013; Frajka-Williams

et al., 2017). Disentangling these modes of internal variability from forced changes is hence important for detection and attribution studies of anthropogenic climate change (Hegerl and Zwiers, 2011; Bindoff et al., 2013; Deser et al., 2020). Indeed, these SST patterns have played a major role in the search for the origin of the most recent global mean surface temperature trend slowdown (Kosaka and Xie, 2013; England et al., 2014). They may also reflect a memory component of the climate system, which can provide enhanced skill in long-term climate predictions (Zhang et al., 2019a). On a broader perspective,

societies are impacted significantly by these SST patterns through associated changes in temperature extremes (e.g. Ruprich-Robert et al. (2018), droughts (e.g. McCabe and Palecki (2006); Delworth et al. (2015)), hurricanes (e.g. Zhang and Delworth (2006)), precipitation patterns (Sutton and Hodson, 2005), and ecosystem productivity (Mantua et al., 1997). (Zhang et al., 2019b) reviews the climate impacts of the Atlantic Multidecadal Variability.

To determine patterns of SST variability, scalar indices measuring average SST anomalies over one or more regions are

often regressed on the global SST field (Deser et al., 2010). Both the North Atlantic and North Pacific are known for their low frequency SST variability and associated pattern. The Atlantic mode, indicated by the average North Atlantic SST, was described by Kushnir (1994) and named the Atlantic Multidecadal Oscillation by Kerr (2000), but in the absence of a single spectral peak, we use the more modern naming convention of Atlantic Multidecadal Variability (AMV). Low frequency Pacific variability was revealed through principal component analysis of North Pacific SSTs (Pacific Decadal Oscillation; Mantua et al.

(1997)) and of the global SSTs (Interdecadal Pacific Oscillation; Power et al. (1999)). The Pacific Decadal Oscillation (PDO) is thought to arise as a combination of low frequency tropical variability and local air-sea interaction (Newman et al., 2016). In the Southern Ocean, SST observations are relatively sparse prior to satellite observations, but signatures of multidecadal variability are found, for example, in the Antarctic sea ice extent (Simpkins et al., 2013). Southern Ocean variability is assessed here through an SST index in the Atlantic sector of the Southern Ocean (Le Bars et al., 2016). This Southern Ocean Mode

(SOM) is a mode of multidecadal variability and is presumably the result of eddy-mean flow interaction (Hogg and Blundell, 2006; Jüling et al., 2018).

Large-scale SST variability is tightly coupled to the Earth's energy balance through surface heat fluxes (SHF) which change the ocean heat content (OHC) and affect the global mean surface temperature (GMST). The SST–SHF coupling is complex: generally, anomalously high SSTs lead to a loss of heat from the oceans, but there is substantial geographic and seasonal

variability depending on atmospheric conditions such as wind speed and cloud presence (Park et al., 2005). In the mid-latitude North Atlantic, sub-decadal atmospheric variability drives the SHF, but low frequency changes in SST drive the SHF on time scales longer than 10 years (Gulev et al., 2013). There are qualitative differences in this air-sea interaction feedback between non-eddying and strongly eddying coupled climate models where the latter match the observed behaviour much better (Small et al., 2020). The OHC is changed by both the SHF and heat divergence (Roberts et al., 2017) and multidecadal OHC changes

are related to the three SST modes discusses here. In the Atlantic, observed OHC changes are strongly modulated by the AMV (Häkkinen et al., 2015). In the Pacific, the PDO SST pattern is replicated in the upper 300 m vertically integrated OHC (Kumar and Wen, 2016). And in the Atlantic sector of the Southern Ocean, the SOM was first described as a mode of OHC variability

with an associated SHF pattern (Le Bars et al., 2016). Furthermore, the global mean SST signal makes up a large part of the GMST signal.

The importance of internal ocean variability in low-frequency climate variations has become more clear in recent years by comparing ocean model results forced by either climatological or observed atmospheric forcing (Penduff et al., 2014). Multidecadal SST variability can arise from a range of processes (Dijkstra, 2013). Energy can be shifted to lower frequencies by integration of high-frequency variability in one component by a slower component of a system, such as the coupled atmosphere-ocean system (Hasselmann, 1976). It can also arise from internal variability in the atmosphere which is imprinted through heat

fluxes on the upper ocean (Eden and Jung, 2001). Internal multidecadal variability in the ocean can arise through specific instabilities, such as the thermal Rossby wave mechanism (Te Raa and Dijkstra, 2002). Results of idealized models have indicated that patterns of internal multidecadal SST variability may arise through noise excited (non)-normal modes (Frankcombe et al., 2009; Weijer and van Sebille, 2014; Dijkstra, 2016) or due to a collective interaction of such modes (Berloff et al., 2007b; Hogg and Blundell, 2006). Such collective interactions can change the vertical heat transport, affecting the stratification which

in turn influences the response of the mixed layer to heat fluxes and/or wind stress (Manucharyan et al., 2017) or impact deep convection at multidecadal timescales (Dufour et al., 2017). Mesoscale eddies can also interact with the mean flow, for example through rectification, which can lead to changes in momentum and vorticity input by wind stress (Hogg and Blundell, 2006; Berloff et al., 2007a). Last, nonlinear advection of kinetic energy in the western boundary current separation region can lead to an inverse temporal cascade in which spectral energy is shifted from high to low frequencies (Martin et al., 2019).

Models participating in phase 5 of the Coupled Model Intercomparison Project (CMIP5; Taylor et al. (2012)) generally underestimate multidecadal climate variability (Cheung et al., 2017; Mann et al., 2020). The amplitudes of Pacific and Atlantic multidecadal variability, as measured by the AMV and PDO indices, are significantly lower than in observations. Further, the regression patterns partially mismatch those in observations, in particular in the Pacific and the western boundary current regions (Kajtar et al., 2019). Hence, CMIP5 models may miss crucial physical processes causing the patterns of multidecadal

variability. The resolution of many, if not all, CMIP5 models is too coarse to capture all relevant internal variability. In at least one ocean model, multidecadal variability appears in a high resolution (strongly eddying) setup but is absent at a lower resolution (non-eddying) version (Le Bars et al., 2016).

Decreasing the horizontal ocean grid spacing from 1° in CMIP5 style GCMs to 0.1° enables new physics with the explicit representation of mesoscale phenomena (Tulloch et al., 2011; Hallberg, 2013). Coherent mesoscale eddies range from 50

to 200 km in size and on a 0.1° grid the Rossby radius is resolved equatorward of 50° latitude (Griffies, 2014). Mesoscale variability arises due to barotropic and baroclinic instability and affects the ocean state in myriad ways (McWilliams, 2008), for example by changing the advection of tracers (in particular affecting the heat and salinity budgets), affecting (re-)stratification (Couvelard et al., 2015; Dufour et al., 2017), and by modifying the mean flow through rectification. Mesoscale eddies also form an integral part of the turbulent energy cascade connecting large-scale potential and kinetic energy input to small-scale

dissipation. Currents, such as narrow western boundary currents, are also much better represented in high resolution simulations as are inter-ocean exchanges of water masses, most notably Agulhas rings (Biastoch et al., 2008).

In most CMIP5 model simulations, mesoscale eddy effects are parametrized with the isopycnal slope mixing parametrization of Gent and McWilliams (1990) (GM). While GM captures many effects of eddies on the circulation, it stabilizes the ocean state and suppresses low frequency variability (Hallberg and Gnanadesikan, 2006; Viebahn et al., 2019). Furthermore, GM

does not capture rectification effects of the eddies on the mean flow. In idealized non-eddying ocean models, the existence of modes of multidecadal variability depends critically on the prescribed eddy diffusivity (Huck et al., 2014). Increasing the resolution to allow mesoscale eddies, Huck et al. (2014) find that the multidecadal variability persists despite changes in the mean circulation, suggesting that the eddy parametrization may suppress such low-frequency variability if tuned incorrectly. In a comparative study with a suite of coupled GFDL models at different resolutions, Griffies et al. (2015) find a stronger

upward heat transport by the mesoscale eddies which counteracts the general downward heat transport by the mean currents. This enhanced vertical heat transport leads to larger heat fluxes and faster adjustment to forcing (Delworth et al., 2012).

With the availability of multi-century simulations of global climate models with strongly eddying ocean components (Kirtman et al., 2012; van Westen and Dijkstra, 2017), with a typical horizontal grid spacing of 10 km or less, it is timely to investigate how the mesoscale ocean flows affect the patterns of multidecadal variability. It is exactly what we do here in this

paper, using simulations with the Community Earth System Model. The paper is organized as follows: section 2 describes the simulations and the methods of analysis, section 3 presents the results, and section 4 discusses and summarizes the results and their implications.

## 2  Model Simulations and Data Analysis

We analyze results from two multi-century present-day control simulations with the Community Earth System Model version

1.0.4 (CESM, Hurrell et al. (2013)), carried out at the Academic Computing Center in Amsterdam (SURFsara), see e.g., van Westen and Dijkstra (2017). Both control simulations use constant year 2000 atmospheric greenhouse gas concentrations forcing, notably $[CO_2]$ = 367 ppm and $[CH_4]$ = 1760 ppb. The CESM components are CAM5 (Community Atmosphere Model), POP2 (Parallel Ocean Program), CICE (sea ice model), and CLM (Community Land Model) which are coupled by the CESM1 coupler. The high resolution simulation ('HR-CESM') employs a 0.1°ocean horizontal grid spacing on a tripolar

grid, while the low resolution ('LR-CESM') simulation has a 1°ocean horizontal grid spacing with a displaced dipolar grid. The effect of subgrid-scale processes on tracer and momentum transport is captured with a biharmonic diffusion operator in HR-CESM and the GM parameterization in LR-CESM (Gent and McWilliams, 1990). The HR-CESM simulation was initialized from a simulation of several decades provided by the National Center for Atmospheric Research, while the LR-CESM simulation was initialized from a decadally averaged ocean state at year 1000 of a CESM 1.1.2 simulation performed

with the same resolution. CESM 1.1.2 and CESM 1.0.4 exhibit only minor differences in their ocean state with CESM 1.0.4 having slightly higher SSTs (not shown). Table 1 summarizes the important simulation characteristics.

From earlier simulations with the same model components (CCSM3.5: Kirtman et al. (2012) and CESM1: Small et al. (2014)), it is known that the climatology of the higher resolution simulation improves compared to the lower resolution simulation in many aspects. The overall SST biases reduce due to a better representation of boundary currents, ocean upwelling,

**Table 1.** Overview of the two CESM simulations with characteristics and names of their ocean and atmosphere grids.

| name | ocean grid | atmosphere grid | years analyzed |
|------|-----------|-----------------|----------------|
| HR-CESM | 0.1°tripole, 42 levels to 6000 m (`tx0.1v2`) | 0.47°×0.63°(`f05`) | 51-300 |
| LR-CESM | 1°dipole, 60 levels to 5500 m (`gx1v6`) | 0.9°×1.25°(`f09`) | 11-260 |

and air-sea interactions (Small et al., 2014; Chang et al., 2020). Some biases remain, though, such as too warm high latitude SSTs which result in low sea ice extent and sea ice volume biases in the high resolution simulation (Kirtman et al., 2012).

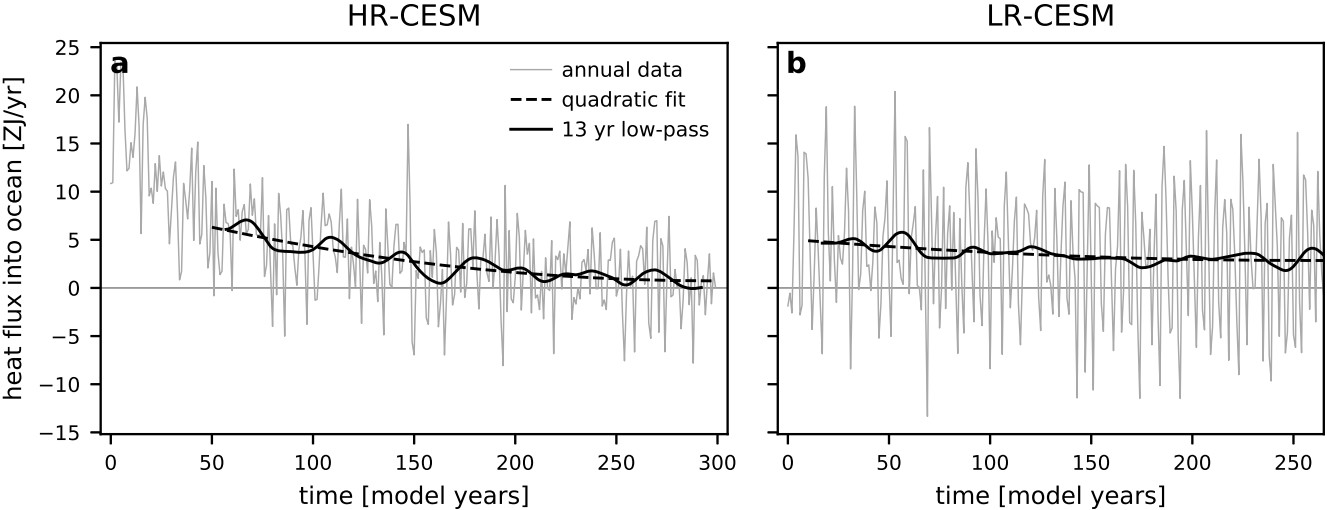

**Figure 1.** The globally averaged surface heat flux into the ocean of the HR- (a) and LR-CESM (b) simulations including the annual time series (thin), the quadratic fit to the 250 analyzed model years (dashed), and the 13 year low-pass filtered version (solid). For the low-pass filtered time series, 7 years are removed at each end to avoid filter edge effects.

Figure 1 shows the global mean surface heat flux into the ocean indicating the equilibration of the simulations. The HR-CESM simulation equilibrates significantly faster than the LR-CESM simulation. The GFDL CM2 climate models show a similarly decreased drift in their higher resolution setups which is due to enhanced upward heat transport by the explicitly 130 resolved eddies (Delworth et al., 2012; Griffies et al., 2015). In our CESM simulations, all basins are initiated too cool and hence the temperature increases at all depths, except for the deep Pacific and some layers at intermediate depths in the Indian Ocean which cool (not shown). On interannual time scales, the Pacific surface heat fluxes vary strongest in comparison with those in the other basins. In particular, in LR-CESM a too strong El Niño-Southern Oscillation (ENSO) signal dominates the global interannual surface heat flux signal (Fig. 1b; Wieners et al. (2019)).

The results in Fig. 1 indicate that a trend must be removed despite the constant forcing as the model simulations are still adjusting towards their statistical equilibrium. We remove a quadratic trend at each grid point of the relevant model output. This approach allows for different proximities to the statistical equilibrium and different time scales of adjustment at each grid

cell. A further choice in the analysis is the selection of model years which are used in the analysis. We ignore the first 50 HR-CESM spinup years as strong adjustment is evident in the global mean SST approximately until year 40. As mentioned, the LR-CESM simulation is initialized at year 1000 of a CESM 1.1.2 simulation with the same forcing and grid that maintains a near-constant surface heat flux of 2.2 ZJ yr$^{-1}$ concurrent to the LR-CESM simulation. We choose to discard the first 10 years of data to avoid fast adjustment processes and analyze years 11-260 (cf. Table 1). Monthly data is deseasonalized by removing the mean seasonal cycle.

To compare the SST model results with observations, we use the HadISST sea surface temperature dataset from 1870-2018 which is provided on a 1°×1°grid (Rayner et al., 2003). As alternative SST observations we use the COBE-SST2 (Hirahara et al., 2014) and ERSSTv5 (Huang et al., 2017) datasets which we limit to the same 149 year period as HadISST. The differences in results with respect to HadISST are discussed in in the Appendix 5. The observational datasets must be detrended with an appropriate estimate of the historical forced signal to allow for a fair comparison with the results of the model simulations, which are obtained under constant forcing. A simple linear detrending is unable to remove the non-linear historical forcing signal (Steinman et al., 2015). When an ensemble of model simulations is available, the forced signal can be approximated as the ensemble mean assuming that the internal variability of the individual ensemble members is uncorrelated. One can use either the mean of a single model ensemble (such as that of the Max-Planck Institute Grand Ensemble (Maher et al., 2019) or the CESM Large Ensemble (Kay et al., 2015)) or a multi-model ensemble (multi-model mean: MMM; such as that of the CMIP5 ensemble). The MMM is generally superior to the single model ensemble mean in the historical period (Frankcombe et al., 2018), so we use CMIP5 ensemble results. The forced signal can furthermore be estimated as a single time series (single factor detrending; used for example by Steinman et al. (2015)) or as a linear combination of different forcing signals (multi-factor detrending as described below).

For the three observational datasets, we choose to separate the natural and anthropogenic forcing signals which was found to be superior to single-factor detrending (Frankcombe et al., 2018). We use the CMIP5 MMM forced with observed aerosol and greenhouse gas concentrations until 2005 and those from the RCP8.5 scenario afterward until 2018 (like e.g. Kajtar et al. (2019)). The natural forcing is assumed to equal the CMIP5 MMM of the historical (natural forcing only) simulations. The anthropogenic contribution ($\mathrm{MMM}_{\mathrm{ant}}$) due to greenhouse gases and aerosols is calculated as the difference between the fully forced MMM ($\mathrm{MMM}_{\mathrm{all}}$) and the MMM with only natural forcing ($\mathrm{MMM}_{\mathrm{nat}}$), i.e.,

$$\mathrm{MMM}_{\mathrm{ant}} = \mathrm{MMM}_{\mathrm{all}} - \mathrm{MMM}_{\mathrm{nat}} \tag{1}$$

At each grid point, scaled versions of these two time series are subtracted from the historical reconstructed SST, such that the detrended SST is

$$\mathrm{SST}_{\mathrm{detr}}(x,y,t) = \mathrm{SST}(x,y,t) - \alpha_1(x,y)\mathrm{MMM}_{\mathrm{nat}}(t) - \alpha_2(x,y)\mathrm{MMM}_{\mathrm{ant}}(t) - \alpha_3(x,y) \tag{2}$$

where the coefficients $\alpha_i$ at each location are determined through multiple linear regression. Stolpe et al. (2017) show for the AMV index and the index of Pacific multidecadal variability by Henley et al. (2015) (which does not equal the PDO index, but shares characteristics) that the various detrending methods result in similar behavior.

From the deseasonalized and detrended model and observational SST fields, we determine area averages and principle components as indices of modes of multidecadal variability. The AMV index is calculated here over the domain [0°N,60°N]×[80°W,0°E], very similar to that in Stolpe et al. (2017) who use the zonal extent [75°W,5°W]. The PDO is captured by the first principal component of the Pacific SSTs north of 20°N as originally proposed by Mantua et al. (1997). The Southern Ocean Mode (SOM) index, proposed by Le Bars et al. (2016) as a signature of the multidecadal variability in the Atlantic sector of the Southern Ocean, is computed over the region [50°S,35°S]×[50°W,0°E]. The index regions are outlined in the respective regression maps of Fig. 3. The indices are defined as the 13 year, second-order Butterworth low-pass filtered monthly time series with the first and last 7 years removed to avoid filter edge effects.

To assess the spatial expression of the modes of multidecadal variability, we use regression plots (Deser et al., 2010). The plotted regression values R are defined as the covariance of the SST field and the normalized index $X(t)$, where $X$ = AMV, PDO, or SOM, according to

$$R(x,y) = \frac{\text{cov}(\text{SST}(x,y,t), X(t))}{\text{std}(X(t))}, \tag{3}$$

such that R has units of temperature. The significance against a no-correlation null hypothesis is tested with a two-tailed Student's t-test. Because the time series are autocorrelated and filtered, the effective number of data points $n'$ is lower than the original sample size $n$ and can be estimated as the maximum of the reduced sample size due to filtering and due to autocorrelation (Trenberth, 1984):

$$n' = n \times \max\left(f\Delta t, \frac{1 - r_{1,X} r_{1,Y}}{1 + r_{1,X} r_{1,Y}}\right), \tag{4}$$

where $r_{1,X}$ $(r_{1,Y})$ is the lag-1 autocorrelation of time series $X$ $(Y)$, $f$ the filtering frequency, and $\Delta t$ the time step.

We analyze time series in the spectral domain with multi-taper spectra (Ghil et al., 2002). This estimator of spectral density is superior to the classic periodogram in that it reduces spectral leakage and is statistically robust, i.e. the estimated noise reduces with more data points. However, as a trade-off the effective spectral resolution is reduced which becomes problematic at periodicities near the length of the time series. We therefore limit our analysis to periodicities below 50 years when focussing on model-observation comparisons but extend the range to 100 year periods for comparisons between the simulations. We use a bandwidth parameter of 2 and the number of tapers is 3. To test significance against a red noise null hypothesis (Hasselmann, 1976), we generate 10,000 Monte-Carlo first-order autoregressive (AR(1)) processes. The autocorrelation coefficient and noise amplitude are estimated with the maximum likelihood estimator via Kalman filtering from the deseasonalized and detrended but unfiltered monthly time series (Durbin and Koopman, 2012).

Beyond analyzing spectral peaks above red noise null hypotheses, we quantify aspects of the spectra by calculating the mean over the multidecadal and interannual time scales as well as fitting a multidecadal spectral slope (Dee et al., 2017; Parsons et al., 2017). We define multidecal variability ('MV') and interannual to sub-decadal variability ('IV') with periods between 10 and 50 years and 2 and 10 years, respectively. To avoid overfitting to higher frequencies when calculating the means and slope, we use weighted averaging and linear regression using the negative logarithm of the frequencies (and not the estimated uncertainty at each frequency). We calculate the standard deviation of the spectral mean based on the 95% jackknife confidence interval. We

assume normally distributed uncertainties in the logarithmic spectral power space and perform a weighted root-mean square addition of the uncertainties in the frequency band implicitly assuming independence. For the weighted linear regression we report the standard error. The MV mean spectral power $\mu_{MV}$ serves as a direct comparison between spectra, while the MV/IV ratio $\mu_{MV}/\mu_{IV}$ speaks to the relative strength of variability between these two frequency windows. The spectral slope indicates the relative strength of low frequency spectral power to high frequency decadal power within the MV window. A negative spectral slope is called 'red' with enhanced spectral power at low frequencies, a positive slope 'blue' with less power at low frequencies, and a near-zero slope 'white' with approximately equal power at all frequencies.

We define the zonally and vertically integrated OHC anomaly, $\mathrm{OHC}_v$, as

$$\mathrm{OHC}_v(y,t) = c_p \int \int \rho\theta(x,y,z,t) \; \mathrm{d}x\,\mathrm{d}z \; , \tag{5}$$

where $\theta$ is the potential temperature, $\rho$ the density, and $c_p = 3996$ J kg$^{-1}$ K$^{-1}$ the seawater heat capacity used in the CESM. The horizontally integrated OHC anomaly $\mathrm{OHC}_h(z,t)$ is

$$\mathrm{OHC}_h(z,t) = c_p \int \int \rho\theta(x,y,z,t) \; \mathrm{d}x\,\mathrm{d}y, \tag{6}$$

The HR-CESM temperature data has been interpolated to a 0.4°rectangular grid and the OHC calculations were performed on that grid. This interpolation introduces very little error and enables the calculation of zonal integrals. The LR-CESM OHC is calculated on the original displaced dipole grid and zonal integrals are performed along the grid x-direction (so not along parallels in the high northern latitudes).

## 3 Results

The results are divided into four subsections: the first describes the variability in SST by means of the chosen indices, the second focuses on the surface heat fluxes (SHF), the third explores the spatial structure of multidecadal variability of the Ocean Heat Content (OHC), and the fourth shows the consequences for the Global Mean Surface Temperature (GMST).

### 3.1 Sea Surface Temperature

Figure 2 shows the AMV, PDO, and SOM indices which all display variability at multidecadal time scales (Fig. 2a-c). The AMV and SOM indices (in units of Kelvin) exhibit a smaller amplitude in the simulations than in the historical data (Fig. 2d). The standard deviations are calculated via the stationary bootstrapping method to avoid biases due to the different time series lengths (Politis and Romano, 1994). Both AMV and SOM amplitudes are higher in HR-CESM than in LR-CESM. The unfiltered monthly PDO time series has unit standard deviation by construction. The PDO amplitude after low-pass filtering (Fig. 2d) is larger in the observations than in the models. The COBE and ERSST standard deviation estimates are included in Fig. 2d and the time series are only shown in Fig. 9. Prior to the satellite era there are discrepancies between the observational SST indices, especially for the SOM index where the sparse data and different data integration approaches result in relatively large differences (Fig. 9).

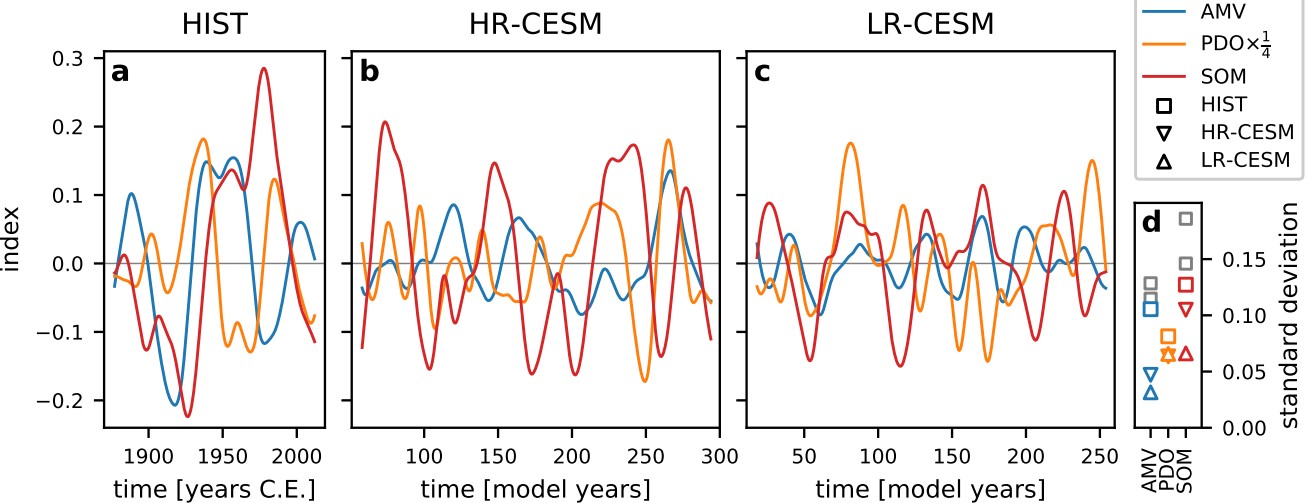

**Figure 2.** The three 13 year low-pass filtered indices of multidecadal variability for the two-factor detrended HadISST data (a) and the quadratically detrended HR- (b) and LR-CESM (c) simulations. Atlantic Multidecadal Variability (blue) and Southern Ocean Mode (red) indices are in units of Kelvin. The monthly time series of the Pacific Decadal Oscillation index (orange) is dimensionless and has been scaled for a comparable amplitude to the AMV and SOM indices. At each end of the time series, 7 years are removed to avoid filter edge effects. Panel (d) shows the standard deviation of the low-pass filtered time series derived via stationary bootstrapping. Grey squares indicate the alternative observational SST products, COBE-SST2 and ERSSTv5. For the PDO all three are equal, for the AMV and SOM, ERSSTv5 exhibits the highest standard deviation.

Figure 3 shows the regression patterns of the detrended SST on the AMV, PDO, and SOM indices for the detrended historical
data (HIST) and the two simulations (HR- and LR-CESM). For the AMV index, the regression pattern of the detrended
historical SST data shows a horse-shoe shape with significant positive regression values throughout the North Atlantic, with a
maximum in the subpolar gyre and a secondary maximum in the subtropical gyre. This is also the case for the COBE dataset
while the two maxima are of the same magnitude in the ERSST data (Fig. 9). The regression patterns of both HR- and LR-
CESM simulations show a comparable pattern to the historical data, albeit with lower regression values. Like the HadISST and
COBE observations, HR-CESM exhibits the strongest regression values in the subpolar gyre, while LR-CESM shows more
pronounced regression values in the subtropical gyre. LR-CESM exhibits significant negative correlations inside the horseshoe
pattern that are not present in the historical observations or HR-CESM. In the historical data and the HR-CESM simulation
regression values are smaller in magnitude outside of the North Atlantic, but the LR-CESM simulation's tropical Pacific
exhibits positive correlation values exceeding those in the North Atlantic. Both simulations show a significant ENSO/PDO-like
regression pattern in the Pacific and dipole structure in the Indian Ocean, whereas the historical data shows only largely non-
significant positive correlations in the tropical Pacific and positive correlations throughout the northern Indian Ocean. Overall,
the areas of significant correlations are larger in LR-CESM than in both the HR-CESM simulation and the historical data.
This suggests possible teleconnections between the Indian and Pacific basins and the Atlantic basin at multidecadal time scales

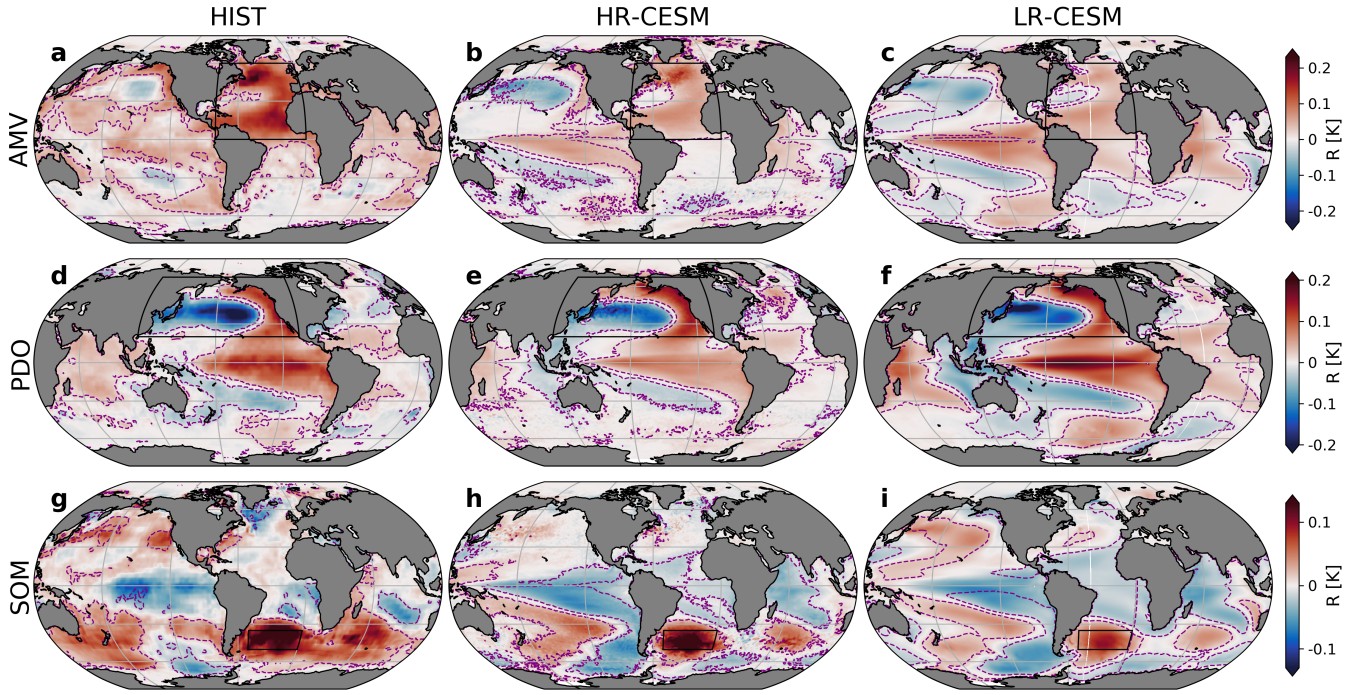

**Figure 3.** Regression maps (values of $R(x, y)$ as in eqn. 3 in Kelvin) of the detrended SST fields onto the AMV (a-c), PDO (d-f), and SOM (g-i) indices for the 149 year detrended HadISST historical dataset (left) and the 250 year data sets of HR- (center) and LR-CESM (right) simulations. Purple dashed lines demarcate the areas of significant correlations at the 98% level. Black boxes outline the index areas. Parallels and meridians are drawn every 30° and 60°, respectively. Note the different colorbar ranges.

especially in LR-CESM, but such correlations are not significant in observations of North Pacific and North Atlantic SSTs (Steinman et al., 2015). The insignificance of the correlations in the observations may be a sign of absent teleconnections, but could also be due to a host of other reasons. These include, for example, sparse early observations, the signal being removed by the (necessarily imperfect) detrending approach, or a non-stationary nature of teleconnections, as suggested for tropical modes variability by Cai et al. (2019).

The PDO regression pattern is characterized by negative values in the Kuroshio extension and positive values in an enclosing horseshoe pattern to the East. The three observational datasets show very similar patterns where the minimum is located east of the dateline and significant positive values exist in the Eastern tropical Pacific (cf. Figs. 3 and 9). Weaker negative (positive) regression values exist in the southwest Pacific (the Bellingshausen Sea). The simulations show largely similar patterns, although the minima in the Kuroshio extension area are shifted west of the dateline, especially so in LR-CESM. The historical data shows uniformly positive correlations in the Indian Ocean where both simulations display a dipole pattern which is more pronounced in LR-CESM. All PDO patterns exhibit positive values in the tropical Pacific, but the LR-CESM regression values are closer to those of the historical data. In LR-CESM, a large area of significant correlations in the tropical Atlantic exists that is absent in both the observations and HR-CESM.

The SOM index regression pattern shows a coherent pattern spanning much of the Southern Ocean at significant levels in the HadISST data with maximum values in the SOM region and a secondary maximum north of the Kerguelen Plateau in the West Indian Ocean sector of the Southern Ocean. In both COBE and ERSST data, the areas of significant correlations are smaller (cf. Figs. 3 and 9). Both simulations exhibit these two positive value areas and whereas the HR-CESM simulation's correlation values are closer to those of the historical SST data, both feature a more azonal pattern with negative values in the Eastern Weddell Gyre. All patterns exhibit negative regression values in the tropical Pacific, but only the simulations feature this structure at significant levels and then as part of a larger South Pacific horseshoe shape pattern that extends south to the Bellingshausen Sea; there, slightly negative values occur in the observations as well. The LR-CESM simulation's area of significant correlations extends further into the Northern Hemisphere, especially in the Pacific.

Figure 4 shows the spectral power of the unfiltered monthly AMV, PDO, and SOM index time series. Again, the historical indices cover 149 years while the simulated ones cover 250 years, resulting in different spectral resolutions. We show the spectra up to a period of 50 years, but note that the spectral estimate of the historical indices is less reliable at low frequencies because of the shorter time series length compared to the model estimates. Most previously published spectral analyses of the North Atlantic observed temperatures find significant periodicities around 50-70 years, but almost all remove a linear trend only which distorts the signal (Steinman et al., 2015). Longer proxy data records reveal significant multidecadal variability around the North Atlantic from sources including Greenland ice cores (Chylek et al., 2011), tree rings (Gray et al., 2004), sediments (Knudsen et al., 2011), or a combination of proxies (Delworth and Mann, 2000; Wang et al., 2017). Palaeo reconstructions have also been performed of the PDO (D'Arrigo et al., 2001; MacDonald and Case, 2005; Felis et al., 2010; O'Mara et al., 2019).

In the detrended observations, significant (99%) spectral power for all three indices occurs at periodicities above 40 years. The PDO signal also extends beyond the red noise (95%) at periods of about 20 years. Between the three observational datasets, the PDO spectra are almost identical and the AMV spectra differ slightly in non-significant ways but the SOM spectra are quite different (Fig. 9). The COBE dataset shows more MV spectral power than HadISST with 99%-significant power above 30 years, and the ERRST dataset exhibits more power at all frequencies than the other two datasets with 99%-significant power above 25 years. In HR-CESM, significant spectral power peaks only exist at 40-50 years for the AMV index (>95%), around 13-15 and 20-25 years for the PDO index (>99%), and around 15 years (>95%) and above 40 years (>95%) for the SOM index. The LR-CESM indices exhibit no significant MV spectral power for either the AMV or SOM indices and the red noise null hypothesis cannot be rejected at 95% level, and only the PDO index has a significant spectral peak (95%) at 16 years. The quantitative measures of the spectra in the MV and IV bands include the MV spectral slope $\beta_{MV}$ (purple), the mean MV spectral power $\mu_{MV}$, and the MV/IV mean power ratio $\mu_{MV}/\mu_{IV}$. For all indices, $\mu_{MV}/\mu_{IV}$ is larger in HR- than in LR-CESM and in the case of AMV and SOM closer to the historical indices. both $\mu_{MV}$ and $\mu_{MV}/\mu_{IV}$ of the AMV are much larger and the spectral slope is much redder in the detrended historical data than in the simulations. The PDO historical and HR-CESM $\beta_{MV}$ are red (negative) while it is white (near-zero) in LR-CESM. The SOM $\beta_{MV}$ of HR-CESM is redder than that of LR-CESM and thus closer to the slope of the historical SOM index.

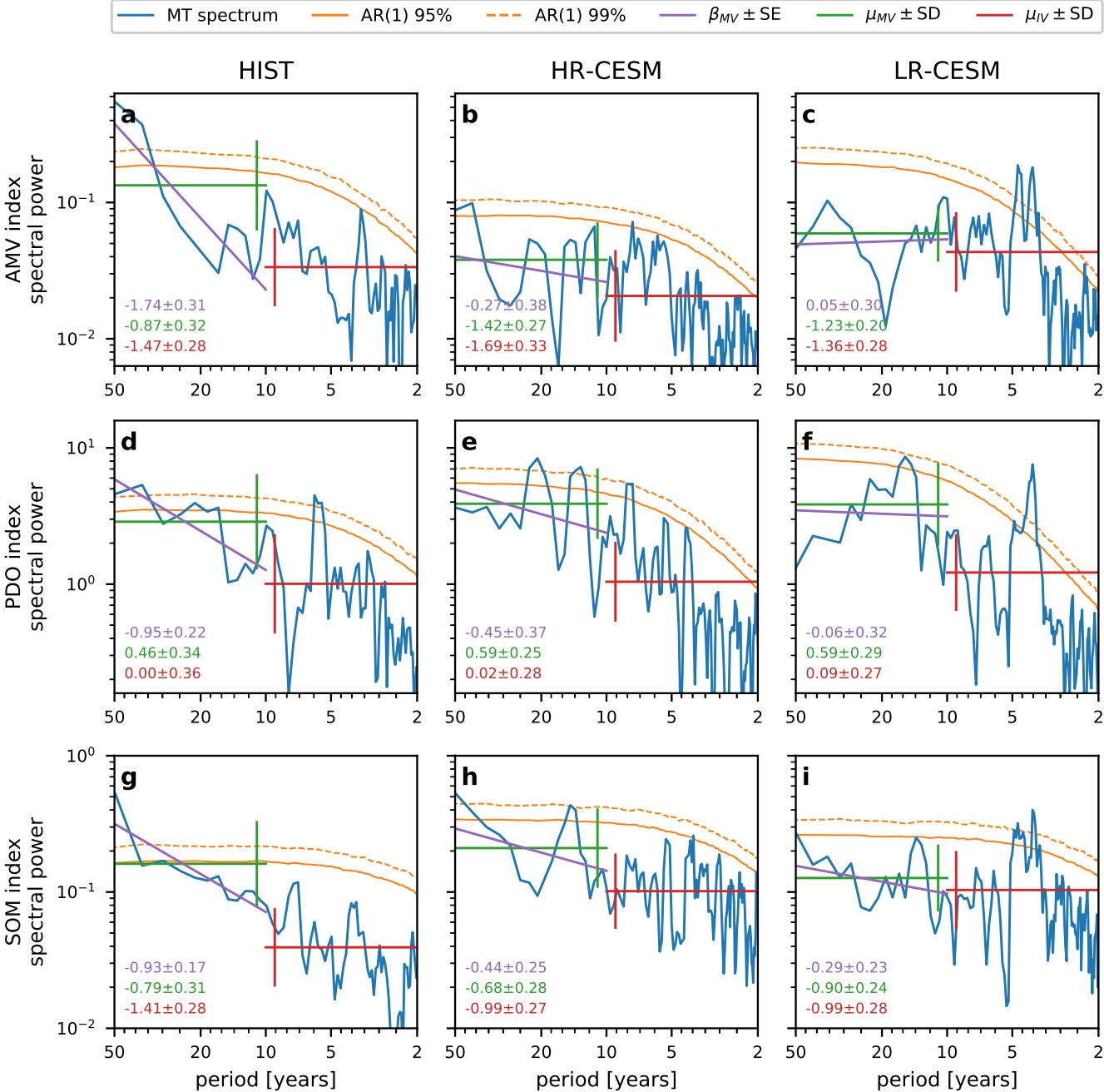

**Figure 4.** Multi-taper spectral estimates of the unfiltered monthly SST indices (blue) underlying the filtered time series of Fig. 2. Panels (a-c) show the AMV, (d-f) the PDO, and (g-i) the SOM index spectra for the two-factor detrended 149 year HadISST dataset (left) and the 250 year HR- and LR-CESM simulations (center and right, respectively). The units of the spectral power are [K$^2$ yr] for the SST average AMV and SOM indices and [yr] for the dimensionless PDO index. As a null hypothesis, 10,000 AR(1) processes were simulated to estimate the 95% and 99% red noise confidence interval (solid and dashed orange lines). The purple line is a linear fit to the spectral slope $\beta$ in the multidecadal variability (MV; 10-50 years) band. The horizontal green and red lines represent the mean spectral power $\mu$ in the MV and the interannual (IV; 2-10 years) bands, respectively. Numbers refer to $\beta_{MV}$ and its standard error (purple) and the decadic logarithm of $\mu MV/IV$ and their standard deviations (green and red).

### 3.2 Surface Heat Fluxes

Figure 5 shows the time series and spectra of the surface heat flux (SHF) into the global ocean as well as into the major ocean basins. The Global Ocean (black) includes all ocean basins and marginal seas, the Atlantic Ocean (blue) is bounded in the north by the Labrador Sea and extends to Iceland, the Pacific (orange) extends up to the Bering Strait, while our Southern Ocean (red) is located south of the parallel at Cape Agulhas at 34°S (inset map in Fig. 5b). For global as well as Atlantic and Pacific heat fluxes, the LR-CESM spectra show enhanced power around 4 and 8 years which is due to a strong and regular ENSO in this simulation (Fig. 5c/d). In the MV band in Fig. 5d, HR-CESM exhibits equal or larger SHF variability than LR-CESM with the exception of the Atlantic. In particular, the Southern Ocean shows enhanced MV spectral power $\mu_{MV}$, and at lower frequencies than the 50 year MV cutoff, all basins and the global SHF exhibit larger spectral power. The $\mu_{MV}/\mu_{IV}$ ratio is larger and the $\beta_{MV}$ are redder globally and for all basins for HR- than for LR-CESM.

### 3.3 Ocean Heat Content

The basin-scale ocean heat content (OHC) changes are largely determined by surface heat fluxes while horizontal heat divergences play a minor role. The strong multidecadal surface heat flux variability motivates the investigation of the structure of OHC anomalies in this section. Long reconstructions of the OHC exist for the industrial period (1870-2015; Zanna et al. (2019)) and even the Common Era (15-2015; Gebbie and Huybers (2019)), but they lack the detail that we aim to investigate here so that we compare model data only.

The first two columns of Figure 6 are Hovmöller diagrams of 13 year low-pass filtered zonally and vertically integrated OHC anomalies, $\mathrm{OHC}_v(y,t)$ (Eq. 5). The last column in Figure 6 shows the standard deviation of the 13 year low-pass filtered anomalies as a function of latitude. Almost everywhere the variability as measured by this standard deviation is higher in HR-CESM than in LR-CESM. The first row shows the global signal which also captures the Southern Ocean signal south of 34°S (marked by the green line). In the HR-CESM Southern Ocean signs of the SOM can be seen with north- and southward propagating anomalies; these are absent in LR-CESM. The Atlantic behavior is qualitatively similar between HR- and LR-CESM with southward propagating anomalies between 40°N and 10°N, although the anomaly amplitude is larger in HR-CESM. Both simulations further show meridionally coherent anomalies south of 10°N in the Atlantic, a pattern seen in observed OHC (Häkkinen et al., 2015). On the other hand, in the Pacific remarkable differences exist: only in the HR-CESM OHC anomaly signals propagate equatorward around 30°N, imprinting on the global pattern as diagonal ridges (compare subplots (g)&(a) with (h)&(b)). This is also visible to a lesser extent in the South Pacific just north of 30°S. In the equatorial Pacific, the unrealistically strong LR-CESM ENSO signal is filtered out by the 13 year low-pass filter. However, compared to the HR-CESM $\mathrm{OHC}_v$, the difference is reduced between the tropical and extratropical low frequency spectral power as revealed by the standard deviations. The Atlantic and Pacific peaks at northern midlatitudes are shifted equatorward in HR-CESM compared to LR-CESM due to a better representation of the western boundary current separation.

Figure 7 shows 13 year low-pass filtered Hovmöller diagrams of the horizontally integrated OHC anomalies, $\mathrm{OHC}_h(z,t)$ (Eq. 6). Globally and in the individual ocean basins, HR-CESM exhibits stronger and deeper OHC variability than LR-CESM.

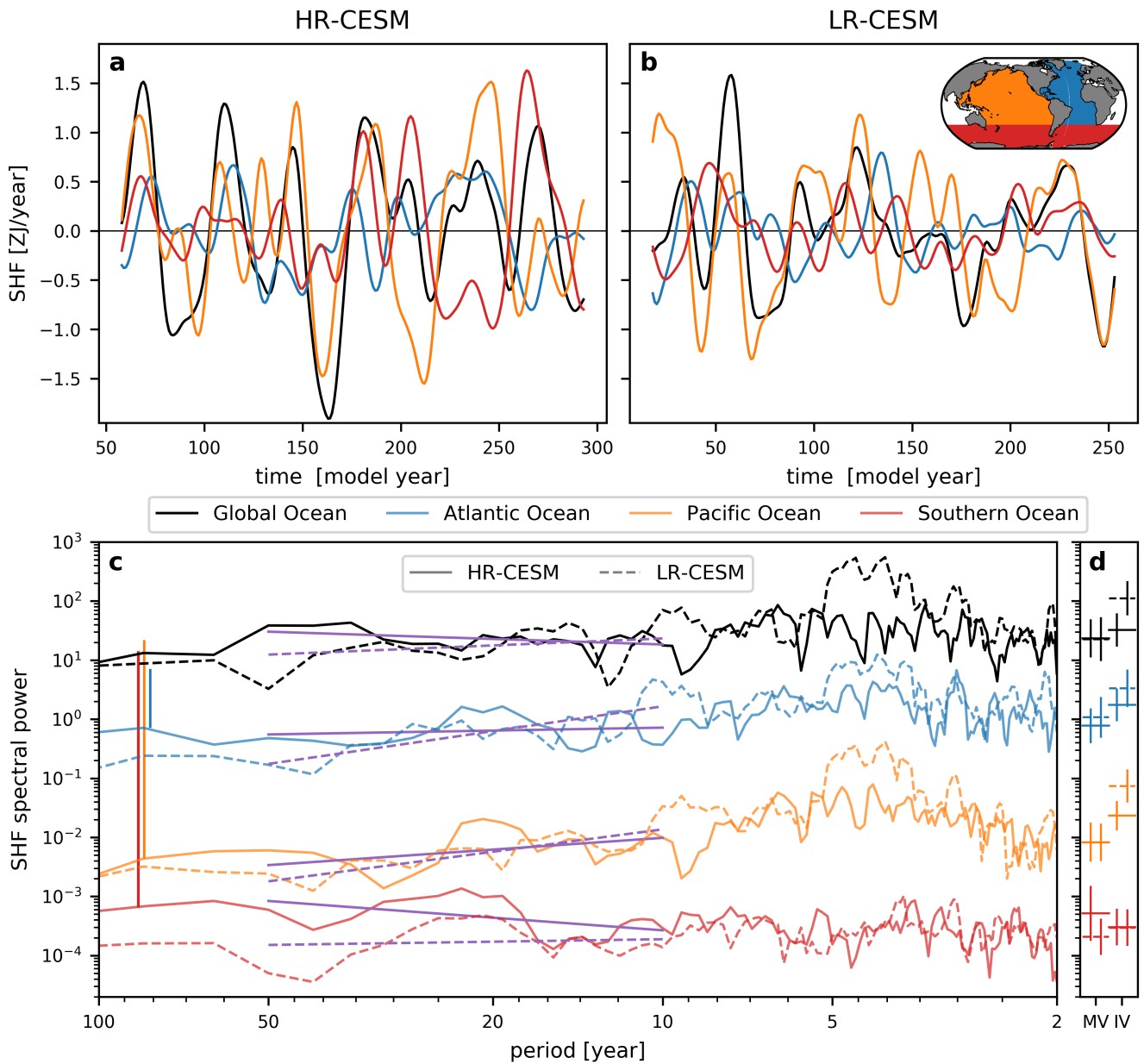

**Figure 5.** Time series for the 13 year low-pass filtered surface heat fluxes (SHF) into the global (black; equals thick solid line of Fig. 1),
Atlantic (blue), Pacific (orange), and Southern (red) oceans for the HR- (a) and LR-CESM (b) simulations. The inset global map defines the
three major ocean basins. Panel (c) shows the multi-taper spectral estimates of the unfiltered, but quadratically detrended time series. The
spectral slopes in the multidecadal variability band (MV; 10-50 years) are shown as purple lines. The spectral power means of the MV and
interannual (IV; 2-10 years) bands are shown in panel (d). The estimated standard deviations are shown as error bars shifted left (right) of
the center for HR-CESM (LR-CESM). For visual clarity, the spectra were separated vertically by multiplying them by constant factors, these
shifts are indicated by vertical lines in (c).

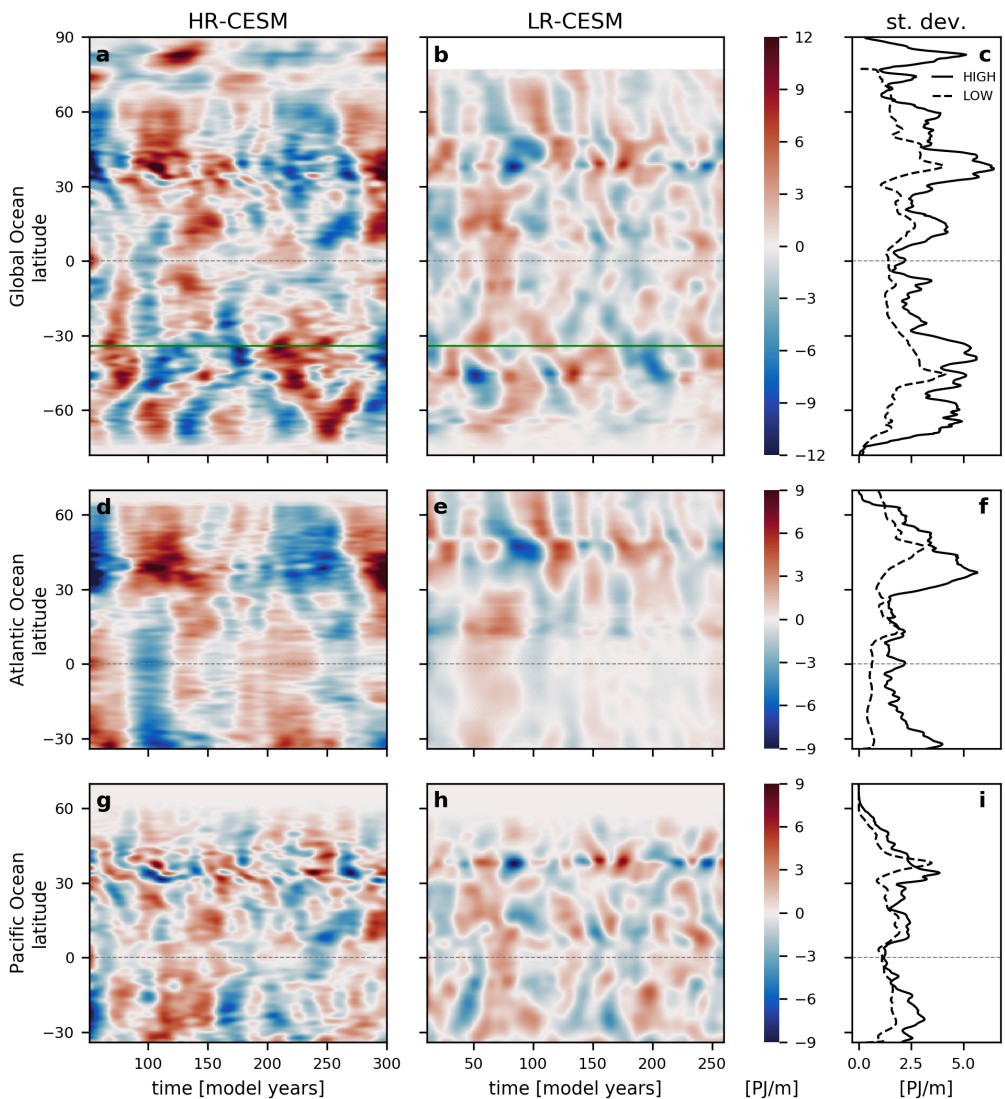

**Figure 6.** The 13 year low-pass filtered zonally and depth-integrated OHC anomalies ($\mathrm{OHC}_v(t,y)$ of Eq. (5)) as Hovmöller diagrams of the HR- (left) and LR-CESM simulations (center), respectively. The top row shows the globally integrated OHC anomaly (which includes the Southern Ocean signal south of 34°S, demarcated with the green line in the first row), while the lower rows show the individual ocean basins. The equator is marked with a thin dashed line. The right column shows the standard deviation in time of the zonally and depth-integrated OHC (solid: HR-CESM, dashed: LR-CESM). The LR-CESM latitudes values are averaged along the grid x-coordinate, such that north of 60°N they are not exactly representing the true latitudes. Note the different color scale ranges of each row with units of $\mathrm{PJ/m} = 10^{15}\,\mathrm{J/m}$.

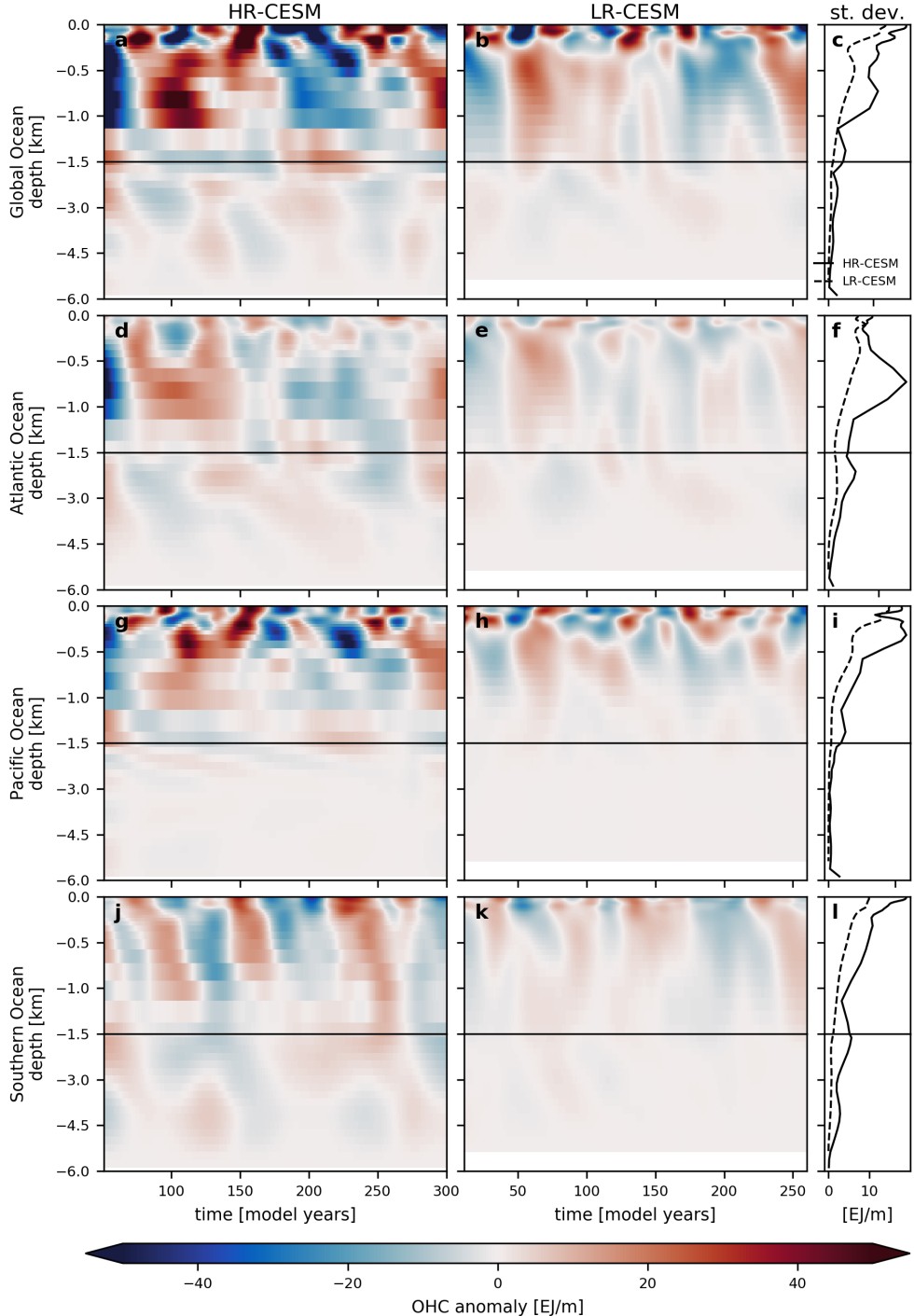

**Figure 7.** The 13 year low-pass filtered horizontally integrated OHC anomalies ($\text{OHC}_h(t,y)$ of Eq. (6)) as Hovmöller diagrams of the HR-(left) and LR-CESM (center) simulations, respectively. The different rows show the global, Atlantic, Pacific, and Southern Ocean integrals over time. Note the change of depth scale at 1500 m. The right column shows the standard deviation in time of horizontally integrated OHC (solid: HR-CESM, dashed: LR-CESM), calculated similarly as in Fig. 6. Units are in $\text{EJ/m} = 10^{18}\,\text{J/m}$.

The global $OHC_h$ anomalies on multidecadal timescales are dominated by those in the upper 1200 m (Fig. 7c). The HR-CESM global multidecadal signal below the mixed layer is made up, in approximately equal parts, of the Atlantic, Pacific, and Southern Ocean components, as they exhibit similar magnitudes in their $OHC_h$ standard deviations. In LR-CESM, the global anomalies are dominated by those in the Pacific. The Atlantic HR-CESM $OHC_h$ anomaly standard deviation shows a pronounced peak between 400 and 1500 m depth. In the Pacific, the variability is larger between 200 and 1500 m in HR-CESM. In the upper 1000 m, the anomalies propagate quickly downwards in the Atlantic but slower in the Pacific. At Pacific depths below 1000 m, a very slow downward propagation of anomalies is visible in HR-CESM that is absent in LR-CESM. The faster vertical propagation of heat anomaly signals in the Atlantic compared to the Pacific can be explained by the presence of the North Atlantic downward branch of the meridional overturning circulation (Buckley and Marshall, 2016). In the Southern Ocean, large differences are evident: the HR-CESM multidecadal variability is much stronger and extends to the full depth as opposed to the LR-CESM variability that is weaker at all depths, in correspondence with results in Le Bars et al. (2016). The SOM index (Fig. 2) correlates very well with Southern Ocean surface $OHC_h$ signal.

### 3.4   Global Mean Surface Temperature

The multidecadal global mean surface temperature (GMST) evolution is a consequence of the heat flux convergence in the atmosphere as energy is exchanged through the sea surface with the oceans and through the top of the atmosphere with outer space. Figure 8 shows the time series and spectral estimates of the GMST. At periods around 4 years, the too strong LR-CESM ENSO (as extensively analyzed in Wieners et al. (2019)), leads to a larger interannual variability in the GMST time series and manifests itself as a peak of the GMST spectral signal. At multidecadal time scales (beyond periods of 13 years), the integrated spectral power in HR-CESM is higher than in LR-CESM, but for periodicities at 35-60 years the LR-CESM GMST spectral power is higher than that of HR-CESM. The spectral power of the two-factor detrended historical GMST data lies between both simulations at sub-decadal time scales, agrees well with both simulations at (inter-) decadal time scales, and exceeds both simulations at periods above 30 years. The MV mean spectral power $\mu_{MV}$ of the HR-CESM is higher than from LR-CESM and closer to the estimate from the detrended historical GMST. The $\mu_{MV}/\mu_{IV}$ ratio of HR-CESM is similar to the historical estimate and both are much larger than for LR-CESM which even exhibits higher IV than MV power. The spectral slope $\beta_{MV}$, however, is less red for HR-CESM than for LR-CESM which is a result of the low LR-CESM spectral power between 16 and 24 years (which is also expressed in the large standard error of the slope estimate).

### 4   Summary and Discussion

We investigated the effect of ocean model resolution on multidecadal variability by contrasting two multi-century simulations with the Community Earth System Model, one with a non-eddying ocean typical of the CMIP5 models ('LR-CESM') and one with a strongly eddying ocean ('HR-CESM'). The enhanced horizontal ocean model resolution allows for more detailed features of the circulation with many documented improvements, such as better representation of boundary currents, sea surface temperature (SST), air-sea exchanges, and internal variability of both high and low frequency relative to the lifetimes of

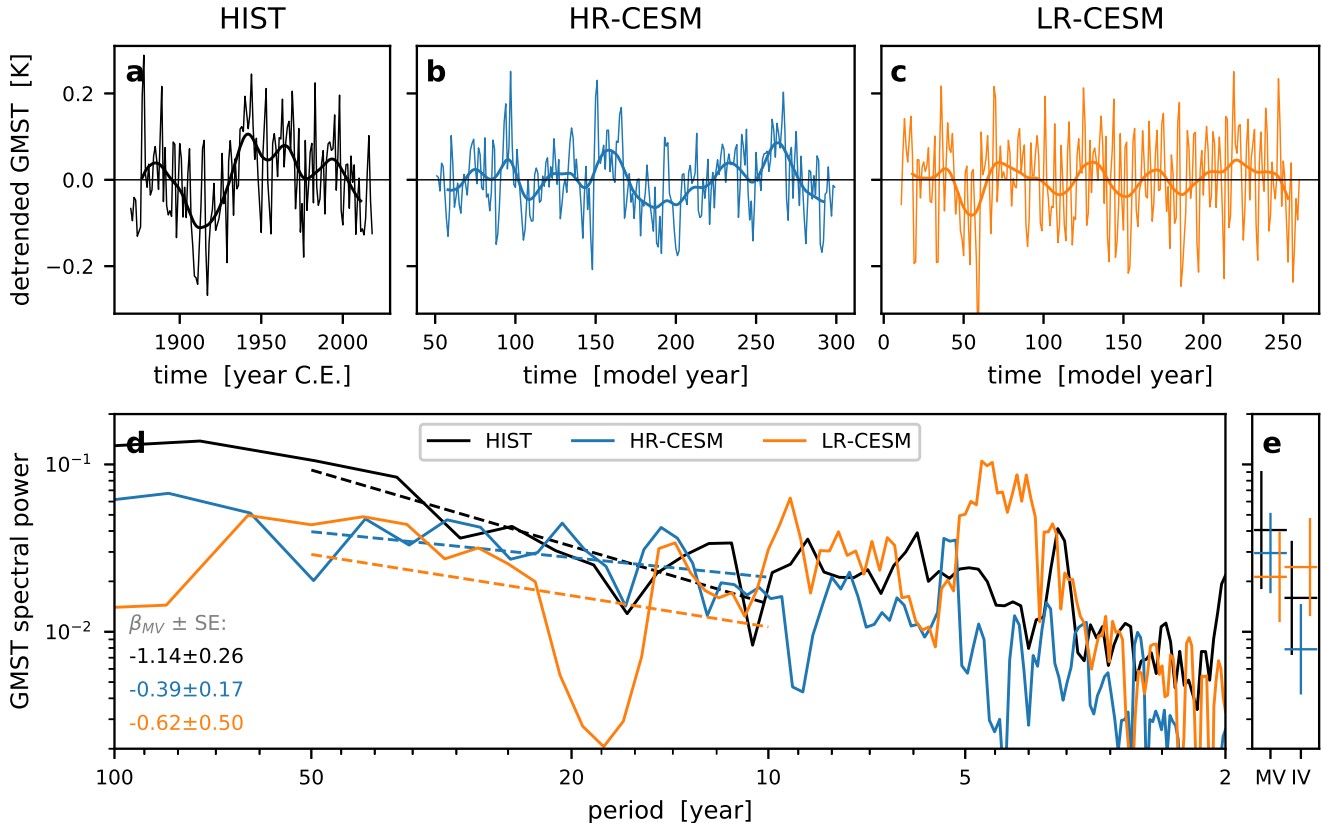

**Figure 8.** Time series of the detrended annual global mean surface temperature (a-c; thin lines) and the 13 year low-pass filtered time series (thick lines). Multi-taper spectral estimates from the time series (d; solid lines) with the fitted spectral slopes $\beta_{MV}$ between 50 and 10 years (dashed). The slope estimates and their standard error written in the lower left corner of panel (c). Panel (e) shows the mean spectral power in the multidecadal (10-50 years) and interannual (2-10 years) bands with the standard deviation estimates as errorbars.

mesoscale ocean eddies. In particular, we focused on three modes of multidecadal SST variability: the Atlantic Multidecadal Variability (AMV), the Pacific Decadal Oscillation (PDO), and the Southern Ocean Mode (SOM). We find that these modes of multidecadal variability are more pronounced in the HR-CESM simulation than in the LR-CESM simulation (Fig. 2) and compare more favourably to observations both in regression patterns (Fig. 3) and spectral properties (Fig. 4). At multidecadal time scales, stronger surface heat fluxes are associated with this larger SST variability (Fig. 5). The integrated ocean heat content (OHC) signal varies more strongly in HR-CESM than in LR-CESM and anomalies extend to greater depths (Figs. 6, 7). However, while the integrated spectral power of the global mean surface temperature (GMST) at multidecadal time scales is larger in HR- than in LR-CESM, there are frequencies at which the LR-CESM GMST variability is larger (Fig. 8). For all analyzed quantities the $\mu_{MV}/\mu_{IV}$ ratio is larger for HR-CESM than for LR-CESM and generally, the spectral slopes $\beta_{MV}$ are redder, and MV mean spectral $\mu_{MV}$ power higher in HR- compared to LR-CESM. As HR-CESM is consistently closer to

observational estimates, we conclude that non-eddying coupled climate models potentially systematically underestimate low frequency variability. From the analyzed HR- and LR-CESM simulations, we conclude that representing mesoscale eddies in the CESM leads to enhanced multidecadal variability.

In the Atlantic, an improvement in the simulation of the AMV pattern is evident in HR-CESM compared to LR-CESM (Fig. 3a-c). While the overall AMV pattern is similar in observations and model simulations, the HR-CESM subpolar maximum in the regression patterns matches the detrended observations in contrast to LR-CESM with its subtropical maximum. Due to the specifics of our detrending approach, the historical regression pattern looks somewhat different from Deser et al. (2010), especially outside the North Atlantic. In particular, the pattern shows a statistically significant positive correlation in
the northwestern tropical Pacific that is not evident in the regression pattern of Deser et al. (2010). The AMV spectra also show that the HR-CESM simulation performs better compared with the historical SST data by showing significant power at multidecadal periodicities against the red noise null hypothesis, while the spectral estimate of the LR-CESM AMV signal fails to reject this null hypothesis (Fig. 4). In CESM Large Ensemble control and historical simulations which uses CESM in the LR-CESM configuration, Kim et al. (2018) also find reduced MV spectral power in the AMV index compared to historical
estimates. While the magnitude of OHC variability is larger in HR-CESM, the meridional and depth structures of the Atlantic OHC variability are similar between HR- and LR-CESM (Figs. 6d-f 7d-f). This suggests that the AMOC effects on the OHC variability are captured with the coarse resolution in agreement with earlier studies (Delworth et al., 1993; Delworth and Mann, 2000) providing support for physical mechanisms of the AMV deduced from idealized models which are independent of mesoscale variability (Te Raa and Dijkstra, 2002). Increasing the resolution can still improve the AMV relative to observations
by reducing ocean mean state biases, in particular the representation of the Gulf Stream and of deep water formation.

We use the PDO index to capture Pacific low frequency variability and the resulting regression patterns are similar between observations and simulations (Fig. 3d-f). The spectral estimates of both the historical PDO and simulated ones extend beyond the 95% confidence interval interdecadal time scales around 20 year periodicity. The LR-CESM simulation exhibits a too strong and regular ENSO signal (Wieners et al. (2019); visible e.g. in Fig. 5c) and the low frequency Pacific $OHC_v$ standard
deviation is relatively larger in the tropics than the extratropics compared to HR-CESM (Fig. 5i). That the PDO partly comprises a low-frequency ENSO signal is visible in the strong tropical Pacific regression maximum in LR-CESM. With the better representation of the western boundary currents in HR-CESM, in particular the Kuroshio, there are qualitative differences in the meridional structure of the Pacific OHC anomaly propagation: around 30°N, and to a lesser degree around 30°S, equatorward propagation is evident in HR-CESM while meridionally coherent anomalies appear in LR-CESM (Fig. 6).

The representation of the Antarctic Circumpolar Current in the Southern Ocean changes dramatically between HR- and LR-CESM due to the presence of mesoscale eddies. The multidecadal SHF variability is much enhanced in the strongly eddying HR-CESM compared to the non-eddying LR-CESM (Fig. 5). Ocean heat anomalies also penetrate much deeper into the Southern Ocean, suggesting a crucial difference between explicitly simulated mesoscale eddies and their parametrization in advecting heat downward (Fig. 7). In HR-CESM, the SOM is visible in the meridional structure with north- and southward
moving anomalies, while anomalies simply converge on 45°S in LR-CESM (Fig. 6). The presence of the SOM is expected in the HR-CESM simulation as it is thought to be caused by the interaction of mesoscale eddies and the mean flow (Hogg and

Blundell, 2006; Le Bars et al., 2016; Jüling et al., 2018). However, the SOM regression patterns of the HR- and LR-CESM simulations are more similar to one another than to the observations with a more azonal, circumpolar wavenumber-3 pattern and stronger correlations extending into the South Pacific (Fig. 3g-i). However, the pattern in the HadISST dataset is likely to be biased due to the limited number of observations in this region, in particular prior to satellite observations.

That strongly eddying climate models should simulate an increase in multidecadal variability relative to higher frequency variability is not immediately obvious as mesoscale eddies have relatively short lifetimes and small spatial scales. Multiple mechanisms through which mesoscale eddies can generate multidecadal variability have been alluded to in the introduction, such as eddy-mean flow interactions (Hogg and Blundell, 2006; Berloff et al., 2007a), the changes to the stratification introducing memory (Manucharyan et al., 2017), or the modification of conditions for convection (Dufour et al., 2017). Furthermore, only eddying models are capable of simulating observed high-frequency variability correctly (Penduff et al., 2011), where an 'inverse temporal cascade' is important for the shift of variance from high to low frequencies. Any mechanism, involving eddies or not, leading to such a cascade thus requires a source of spectral energy at high frequencies which can be provided by the eddies. It is beyond the scope of this study to analyze the exact mechanisms at play in HR-CESM, but it will be the subject of further study.

Naturally, the comparison between observations and model results is fraught with challenges, not least of which is the choice of an appropriate detrending procedure. The inadequate removal of the forced signal in the observations leads to biased regression patterns (Brown et al., 2015). In the model simulations the forcing is held constant and there is only a modest model drift to detrend (Fig. 1) for which we chose to a second order polynomial. The observations are based on the climate system's response to non-stationary forcing and we chose the scaled multi-model mean approach (Frankcombe et al., 2015). Here the detrending signal is derived from the multi-model mean of the CMIP5 models which are biased in their own ways and have different relative climate sensitivities to different forcings. Also, the external forcing prescribed in the CMIP5 models is in itself uncertain and hence may alter the relative contribution due to external forcing and internal variability. Furthermore, the atmospheric grid spacing is refined from 1° in LR-CESM to 0.5° in HR-CESM. However, no new essential atmospheric processes are resolved, so no significant changes are expected apart from coupling to different ocean boundary conditions.

These caveats do not detract from our main point which is to emphasize that low frequency variability may be underestimated in climate models with low-resolution ocean model components and that representing mesoscale eddies in climate models can improve the simulation of this variability. Of course, we showed this here only for the CESM but expect the improvement of multidecadal variability to be generalizable to other coupled climate models, as some of the features like enhanced vertical heat transport are consistent with results obtained with other models (Griffies et al., 2015). The introduction of high frequency variability through mesoscale ocean features and the reduced ocean state bias can further improve the simulations' multidecadal variability skill. Certainly, better eddy parametrizations may improve shortcomings of the traditional Gent-McWilliams approach (Zanna et al., 2017). Recently, Mann et al. (2020) claimed an absence of evidence for multidecadal oscillations in CMIP5 models, defined as spectral peaks above a null hypothesis, but all these models use non-eddying ocean components. As most CMIP6 models outside the High Resolution Model Intercomparison Project (HiResMIP, Haarsma et al. (2016)) still use

non-eddying ocean components, not much improvement is expected on the representation of multidecadal variability in these models compared to CMIP5 models.

Internal variability obscures any forced GMST signal and periods of accelerated and decelerated warming are observed, such as the recent warming trend slowdown (Medhaug et al., 2017). Many studies use global circulation models with coarse resolution ocean components to investigate, for example, the origin of these so-called hiatuses (e.g. Maher et al. (2014)). In light of our findings, estimates of the frequency and magnitude of excursions from the forced trend may be systematically low biased in low-resolution models, as internal multidecadal variability is underestimated. The increased internal variability also implies that the attribution of forced signals becomes more difficult and the issue of the origin of the recent warming trend slowdown may therefore never be satisfactorily resolved (Hedemann et al., 2017). Finally, our finding of stronger than previously modelled multidecadal OHC variability underlines the necessity of continued, long-term observations of the oceans with the ARGO program.

## 5 Appendix

In the main text we use only the HadISST SST dataset. Figure 9 shows the index time series, their spectra, and regression patterns based also on the COBE-SST2 and ERSSTv5 products. All data is based on deseasonalized and two-factor detrended monthly time series in the period 1870-2019. For the time series of all three indices, the agreement improves in time as more observations are available, the time series from the different observational datasets are well correlated from the 1980s when satellite observations became available. The AMV time series and spectra exhibit minor differences, no significant spectral peaks are added or removed. The PDO spectra are almost identical, except that both the COBE and ERSST dataset show 95%-significant spectral power above 25 years where HadISST only shows that above 35 years. The SOM time series and spectra differ the most, as one would expect due to the sparsity of observations prior to the satellite era. The COBE dataset exhibits more MV spectral power than HadISST, and the ERSST dataset exhibits more spectral power than the other datasets in both the interannual and multidecadal bands. As a result also the significance levels are very different and both COBE and ERSST show significant spectral power at higher MV frequencies than HadISST.

The regression patterns are overall similar between the observations, but areas of significant correlations outside the index areas change slightly. The AMV regression pattern of ERSST shows two equal correlation maxima in the North Atlantic while both HadISST and COBE exhibit a stronger maximum in the subpolar gyre. The PDO regression patterns are very similar as expected as the time series and spectra are almost identical as well. The SOM patterns differ the most, but mostly regarding the areas of significant correlations in the Southern Ocean outside the Atlantic sector

*Code and data availability.* The analysis scripts are available at `zenodo link to final code repository`, while the model output is stored at SURFsara and available upon request to the corresponding author. COBE-SST2 data provided by the NOAA/OAR/ESRL PSL, Boulder, Colorado, USA, from their Web site at https://psl.noaa.gov/data/gridded/data.cobe2.html (Hirahara et al. (2014), last ac-

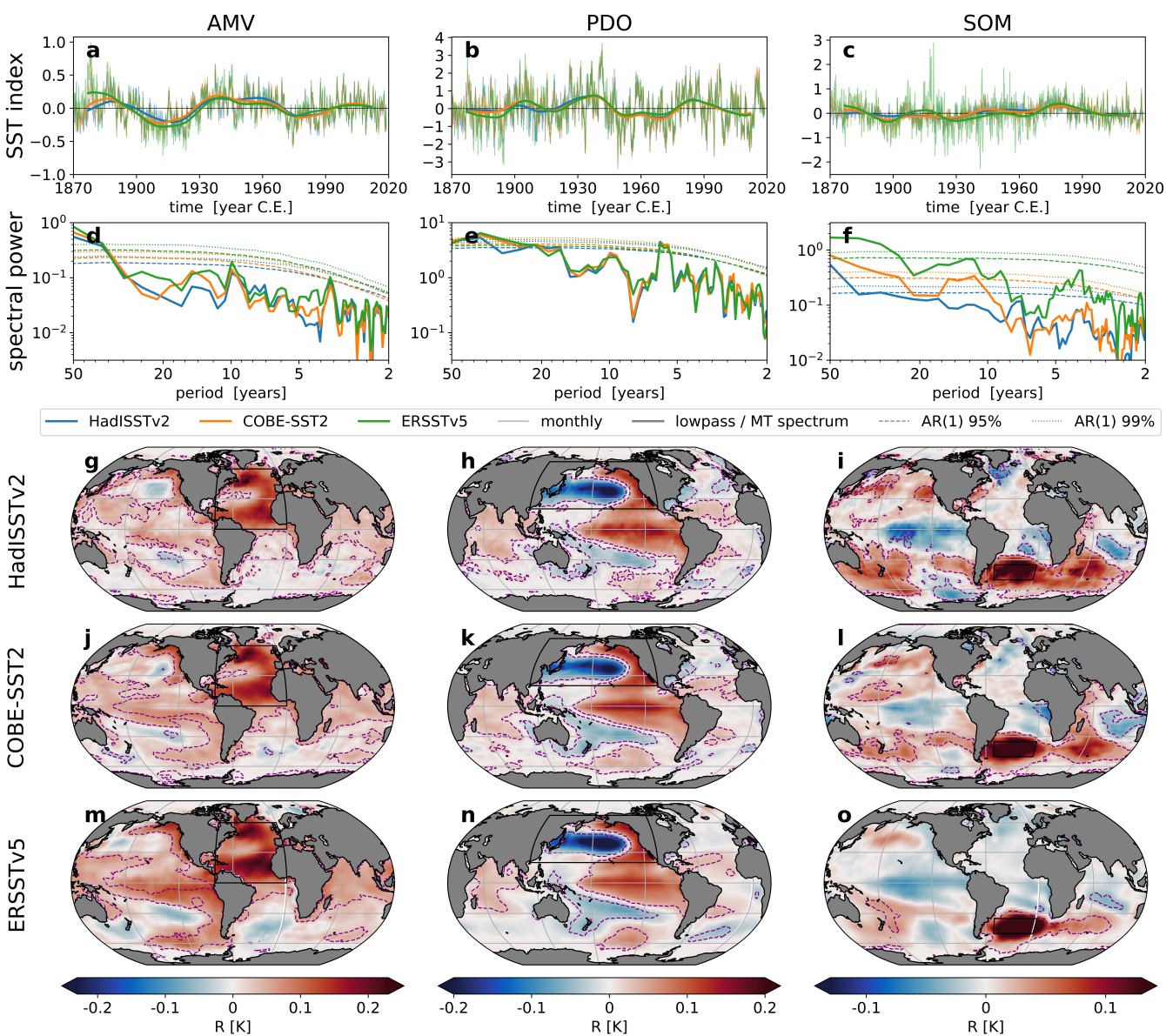

**Figure 9.** Comparing results from three SST products: HadISSTv2 (as in main text), COBE-SST2, and ERSSTv5. Panels (a–c) show the SST indices like Fig. 2, (d–f) the spectra like Fig. 4, and (g–o) regression patterns like Fig. 3. The time series plots show both the monthly (thin) and 13 year lowpass filtered (thick) deseasonalized and two-factor detrended time series. For each of the multitaper spectral estimates, 10,000 AR(1) processes have been simulated to provide a 95% (dashed) and 99% (dotted) significance estimate. The regression maps show the correlation coefficient of the monthly deseasonalized and detrended SST data with indices. The index areas are marked by black rectangles and areas of 98% correlation significance are enclosed by purple dashed lines.

cess 23.05.2021). ERSSTv5 data were obtained from https://www.ncdc.noaa.gov/data-access/marineocean-data/extended-reconstructed-sea-surface-temperature-ersst-v5 (Huang et al. (2017), last access 23.05.2021). HadISST version 2 data were obtained from https://www.metoffice.gov.uk/hado (Rayner et al. (2003), last access 23.05.2021).

*Author contributions.* AJ, AvdH, and HD conceived the presented ideas in this study. AJ performed the analysis and wrote the manuscript. AvdH and HD contributed to writing the paper.

*Competing interests.* The authors declare that they have no conflict of interest.

*Acknowledgements.* This work was carried out under the program of the Netherlands Earth System Science Centre (NESSC), financially supported by the Ministry of Education, Culture and Science (OCW) (Grantnr. 024.002.001). The computations were performed on the Carte-480 sius high performance computer at SURFsara in Amsterdam. Use of the Cartesius computing facilities was sponsored by the Netherlands Science Foundation (NWO) under the project 17239. We thank Michael Kliphuis (IMAU) for carrying out the computations.

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
