# Peer review of "Effects of strongly eddying oceans on multidecadal climate variability in the Community Earth System Model"

_Ocean Science, 2020_

## Referee Comment (RC1) · Anonymous Referee #1 · 5 Oct 2020

This study aimed to test the hypothesis that mesoscale resolving climate models can better simulate multidecadal climate variability of the climate system. To do so, the authors compared two simulations with constant 2000 GHG condition from the Community Earth System Model (CESM) – one with 1 degree and one with 0.1 degree ocean grid – with observational records. Specifically, the authors compared 3 climate indices between these datasets: Atlantic Multidecadal Variability (AMV), Pacific Decadal Oscillation (PDO), and Southern Ocean Mode (SOM). The authors also compared surface heat fluxes (SHF), ocean heat content (OHC), and global mean surface temperature (GMST) between these two simulations. The authors found an improvement in simulating these climate modes using the high resolution model. The low frequency of SHF

and OHC were also generally larger in the high resolution model. The difference in GMST magnitude between the two simulations were timescale dependent. Based on these results, the authors asserted that using mesoscale resolved climate model could improve the representation of multidecadal climate variability in climate models.

Overall, I thought this study was very interesting and significant. In particular, this study tried to resolve issues raised regarding climate models' ability to capture 'low frequency' climate variability by a series of studies (e.g. Laepple and Huybers 2014a, 2014b; Frankcombe et al. 2015; Cheung et al. 2017; Parsons et al. 2017). The approach used in this study also moved past from using idealized model to quantify the effects of small scale processes on large scale circulation. To my knowledge, the analyses done also seemed valid. Though, I believe this study would benefit from further improving the analyses done in this study and the presentation style in certain parts of the manuscript. Detail comments are below.

Major comments:

1) It is a bit unclear what specific timescales the authors are focusing on. Throughout the article, the authors used 'multidecadal' variability without specifying the timescales. This becomes confusing when the authors discussed about specific frequencies. For instance, if high resolution model does a better job in simulating a specific frequency but not some other frequency (where both can be classified as 'multidecadal' timescale), does that mean the high resolution model is doing a better job or does it not?

2) One major aspect of this study is the comparison of data in the frequency domain. In general, I find a little bit difficult to convince myself that the high resolution model is indeed better (or significantly different from) than the low resolution model in simulating multidecadal variability by looking at the power spectrum.

a. For instance, the authors argued that the high resolution model does a better job in simulating the power spectra of all the climate modes by suggesting consistency

between spectral peaks shown in observations and high resolution model. However, there are two problems. First, this assumes that having the correct spectral peak is what we should aim for. However, as suggested in the text, getting the correct spectral peak that is internal to the climate system from observation is difficult. Spectral peaks can arise from incorrectly removing forced signal from observation. So, having similar spectral peaks between observation and model do not necessarily mean the model simulation is better. Second, both high resolution and low resolution simulations seem to have similar spectral power. The main difference between these two simulations is related to the 'null hypothesis', where the low resolution climate model has a higher null threshold. As such, I wonder if comparing spectral peaks is a good way to quantify whether the high resolution simulation is better than the low resolution model.

b. Another example is in SHF and GMST. The effects of reduced variability on ENSO band in the high resolution simulation is undoubtedly pretty clear. However, the difference between high and low resolution simulation becomes a lot less clear on longer timescales (i.e. multidecadal timescales). This again makes comparison on the frequency spectrum less powerful and makes the result less clear. I suggest the authors can try to come up with some metrics to compare the power spectrum more quantitatively. The simplest way to compare agreements between power spectra would be to compute the spectral coherence. Alternatively, multiple studies have tried to compare the spectral slopes over a range of timescale to argue for/against of underestimation of multidecadal variability (e.g. Dee et al. 2016; Parsons et al. 2017). I think these examples can serve as a framework to develop more quantitative comparison between power spectrum.

3) SHF and OHC are undoubtedly related to SST. However, SHF and OHC were not mentioned explicitly throughout the introduction section and did not come up until the first paragraph of results. This introduction of SHF and OHC is rather abrupt, and it is not totally clear why a comparison of SHF and OHC between high and low resolution model is needed in the context of this study. Having more explicit description on the

motivation to analyze SHF and OHC in the context of this study in the introduction part would be helpful.

4) In addition, I am also having a little bit trouble to tie the results in the SHF and OHC sections back to the SST patterns (i.e. AMV, PDO, SOM). Aside from SOM, I don't think there is enough discussion about how the more variable SHF and OHC are related to the more accurate SST patterns for AMV and PDO (or vice versa). I understand that it is outside of the scope of this study to identify the underlying mechanism that leads to an improvement in simulating multidecadal variability, but since this study analyzed SHF and OHC in tandem with these climate modes, I think it's reasonable to at least make more explicit connections between these variables in the discussion section.

5) A large portion of results that described how the calculations were done should be moved to the method section. This includes Lines 173 – 184, Lines 249 – 256, Lines 270 – 271. There should also be discussion on how GMST is defined here.

6) I suggest adding a sentence or two that describes what the implications are based on the results obtained from this study. That way, the readers can understand the major takeaways of this study from the abstract.

Specific comments:

1) Line 23, Zhang et al. (2019) is probably not the best reference for this sentence. There are many other papers that are more relevant and explicit in discussing about the importance to disentangle modes of internal variability from forced changes for detection and attribution studies. Examples include: Hegerl and Zwiers 2011; Bindoff et al. 2013; Deser et al. 2020.

2) Lines 26 – 27, I believe there are other relevant studies that the author should cite, examples include: temperature extremes (e.g. Ruprich-Robert et al. 2018), droughts (e.g. McCabe et al. 2005; Delworth et al. 2015), hurricanes (e.g. Zhang and Delworth 2006). For Atlantic, I think Zhang et al. 2019 (Rev. Geophys.) provides a good overview

of societally relevant impacts induced by AMV.

3) Lines 31 – 32, I'm not sure if Atlantic Multidecadal Oscillation was named in Kushnir (1994). To my knowledge, it is more explicitly mentioned and defined in Kerr (2000), although many studies in the 1990s have already identified multidecadal variations in the North Atlantic.

4) Lines 122 – 123, it is unclear how the data is de-seasonalized. My understanding of how de-seasonalize is commonly done is by first calculating the monthly climatology (Jan – Dec) and then subtracting it from the monthly data (e.g. October 2020 SST – averaged October SST). I think the first part of the sentence described this process, but the 'i.e. . . .' describes something else – annual SST of each year is subtracted from monthly SST of that year. It would be great if this could be clarified.

5) Figure 2d. It should be noted that the standard deviation comparison between observation and models isn't quite 'apples' to 'apples' because their temporal lengths are different. Even though I don't expect the results would change significantly, I would try to bootstrap the results to account for sampling uncertainty.

6) Line 179, please specify which order of Butterworth filter was used.

7) Lines 198 – 199, it's worthy to point out Steinman et al. (2015) only focused on North Atlantic and North Pacific, and they did not look at the relationship between Atlantic, Pacific and Indian together.

8) Lines 218 – 220, please double check the sentence, something is missing. Right now, I have trouble understanding this sentence.

9) Lines 224 – 226, there are also studies that tried to reconstruct decadal variability in the Pacific. Since the authors mentioned paleoclimate studies in the Atlantic, I think it won't be complete without a discussion (or at least mentioning) on paleoclimate studies in the Pacific. Examples include: D'Arrigo et al. 2001; MacDonald and Case 2005; Felis et al. 2010; Lapointe 2017; O'Mara et al. 2019.

10) Figure 5a–b, the unit on the y-axis is wrong – it should be [heat/time] since it's flux data.

11) Lines 239 – 240, which frequencies of the spectral power for Atlantic and Pacific were integrated over?

12) Lines 244 – 245, citation is needed. There are studies that showed ocean dynamics (i.e. horizontal divergence) plays a significant role in driving OHC change (e.g. Roberts et al. 2017; Small et al. 2020).

13) Lines 245 – 246, I think this statement requires clarification. In your previous subsection, you showed that Southern Ocean (and it seems like the global ocean also) SHF exhibit a white noise whereas the Atlantic and Pacific exhibit a blue noise behavior in low frequencies. These results, to me, don't suggest a particularly strong multidecadal surface heat flux variability, at least relative to high frequency variability.

14) Line 249, please explain why and how an interpolation was done here.

15) Figure 6, I'm curious as to why Indian Ocean is included here. Throughout the manuscript, this is the only place where data from the Indian Ocean was analyzed. Even though I think it's interesting to look at it, I'm not sure if it is very relevant to this manuscript, where the target is more on global, Pacific, Atlantic, and the Southern Oceans.

16) Lines 330 – 331, please double check the figure that is referred to. I think it's supposed to be Fig 6i instead of 5i. If so, Fig 6i represents depth and zonally integrated OHC but not SHF.

17) Line 375, note that Mann et al. (2020) showed that CMIP5 do not show multidecadal *oscillations* (they defined it as significant spectral peak against a null hypothesis) but not absence of multidecadal variability.

18) Line 377, 'multidecacal' should be multidecadal.

References:

Anderson, W. B., Seager, R., Baethgen, W., Cane, M., & You, L. (2019). Synchronous crop failures and climate-forced production variability. Science advances, 5(7), eaaw1976.

Bindoff, N. L., Stott, P. A., AchutaRao, K. M., Allen, M. R., Gillett, N., Gutzler, D., ... & Mokhov, I. I. (2013). Detection and attribution of climate change: from global to regional.

d'Arrigo, R., Villalba, R., & Wiles, G. (2001). Tree-ring estimates of Pacific decadal climate variability. Climate Dynamics, 18(3-4), 219-224.

Dee, S. G., Parsons, L. A., Loope, G. R., Overpeck, J. T., Ault, T. R., & Emile-Geay, J. (2017). Improved spectral comparisons of paleoclimate models and observations via proxy system modeling: Implications for multi-decadal variability. Earth and Planetary Science Letters, 476, 34-46.

Delworth, T. L., Zeng, F., Rosati, A., Vecchi, G. A., & Wittenberg, A. T. (2015). A link between the hiatus in global warming and North American drought. Journal of Climate, 28(9), 3834-3845.

Deser, C., Lehner, F., Rodgers, K. B., Ault, T., Delworth, T. L., DiNezio, P. N., ... & Kay, J. E. (2020). Insights from Earth system model initial-condition large ensembles and future prospects. Nature Climate Change, 1-10.

Felis, T., Suzuki, A., Kuhnert, H., Rimbu, N., & Kawahata, H. (2010). Pacific Decadal Oscillation documented in a coral record of North Pacific winter temperature since 1873. Geophysical Research Letters, 37(14).

Hegerl, G., & Zwiers, F. (2011). Use of models in detection and attribution of climate change. Wiley interdisciplinary reviews: climate change, 2(4), 570-591.

Kerr, R. A. (2000). A North Atlantic climate pacemaker for the centuries. Science,

288(5473), 1984-1985.

Lapointe, F., Francus, P., Lamoureux, S. F., Vuille, M., Jenny, J. P., Bradley, R. S., & Massa, C. (2017). Influence of North Pacific decadal variability on the western Canadian Arctic over the past 700 years. Climate of the Past, 13(4), 411-420.

Laepple, T., & Huybers, P. (2014a). Ocean surface temperature variability: Large model–data differences at decadal and longer periods. Proceedings of the National Academy of Sciences, 111(47), 16682-16687.

Laepple, T., & Huybers, P. (2014b). Global and regional variability in marine surface temperatures. Geophysical Research Letters, 41(7), 2528-2534.

MacDonald, G. M., & Case, R. A. (2005). Variations in the Pacific Decadal Oscillation over the past millennium. Geophysical Research Letters, 32(8).

O'Mara, N. A., Cheung, A. H., Kelly, C. S., Sandwick, S., Herbert, T. D., Russell, J. M., ... & Herguera, J. C. (2019). Subtropical Pacific Ocean temperature fluctuations in the Common Era: Multidecadal variability and its relationship with Southwestern North American megadroughts. Geophysical Research Letters, 46(24), 14662-14673.

Parsons, L. A., Loope, G. R., Overpeck, J. T., Ault, T. R., Stouffer, R., & Cole, J. E. (2017). Temperature and precipitation variance in CMIP5 simulations and paleoclimate records of the last millennium. Journal of Climate, 30(22), 8885-8912.

Roberts, C. D., Palmer, M. D., Allan, R. P., Desbruyeres, D. G., Hyder, P., Liu, C., & Smith, D. (2017). Surface flux and ocean heat transport convergence contributions to seasonal and interannual variations of ocean heat content. Journal of Geophysical Research: Oceans, 122(1), 726-744.

Ruprich-Robert, Y., Delworth, T., Msadek, R., Castruccio, F., Yeager, S., & Danabasoglu, G. (2018). Impacts of the Atlantic multidecadal variability on North American summer climate and heat waves. Journal of Climate, 31(9), 3679-3700.

Small, R. J., Bryan, F. O., Bishop, S. P., Larson, S., & Tomas, R. A. (2020). What Drives Upper-Ocean Temperature Variability in Coupled Climate Models and Observations?. Journal of Climate, 33(2), 577-596.

Zhang, R., & Delworth, T. L. (2006). Impact of Atlantic multidecadal oscillations on India/Sahel rainfall and Atlantic hurricanes. Geophysical Research Letters, 33(17).

Zhang, R., Sutton, R., Danabasoglu, G., Kwon, Y. O., Marsh, R., Yeager, S. G., ... & Little, C. M. (2019). A review of the role of the Atlantic meridional overturning circulation in Atlantic multidecadal variability and associated climate impacts. Reviews of Geophysics, 57(2), 316-375.

---

## Referee Comment (RC2) · Anonymous Referee #2 · 19 Oct 2020

This study analyses the impact of model resolution on the simulation of multidecadal climate variability. 250-year simulations are run with the Community Earth System Model at high (0.1° ocean) and low (1° ocean) resolution, which are then compared to 149 years of observational data. It is found that the higher resolution run simulates larger multidecadal variability in the Atlantic and Southern Ocean (and more like observations), with little difference between the two runs in the Northern Pacific. The improvements are linked to better resolution of mesoscale ocean dynamics, and therefore larger heat content variability in the higher resolution run. Some assessment is made with regards to the impact on global mean surface temperature (GMST), but little difference is seen in multidecadal GMST variability across the two resolutions.

[Figure]

Major Points

The paper is well-written, well-presented, and certainly worthy of publication in Ocean Science. The question around the impact of model resolution on the representation of multidecadal variability is likely to be of interest to the community. Clearly a lot of time has been spent on polishing the manuscript, and it is in an excellent state. There are a small number of minor points below that may require some attention.

Minor Points

+ L9: "The effect on global mean surface temperature is relatively minor". It might be better to clarify here that the effect on multidecadal GMST variability is relatively minor, since you show that there are changes to interannual variability.

+ L34: please indicate here that Pacific Decadal Oscillation is abbreviated to PDO later (PDO is used at L58 for the first time).

+ L151: appears to be the first use of 'SOM', and yet to be defined.

+ L173-178: it might be useful to move the index definitions into Section 2.

+ L181: "The AMV and SOM indices (in units of Kelvin) exhibit a smaller amplitude in the simulations than in the historical data". This is also true for the PDO index?

+ L181: How much of the difference between observations and model runs can be attributed to the different data lengths? In panel d, it might be helpful to show uncertainty bars indicating the range of standard deviations for the model data, if you were to compute it in 149-year moving windows (i.e. same length as observed data).

+ L183: "Larger PDO amplitudes...". I don't follow this sentence. Larger PDO amplitudes with respect to what?

+ Fig. 2 caption: The sentence beginning with "The monthly time series of..." requires some editing.

+ L198: "This suggests possible correlations between the Indian and Pacific basins and the Atlantic basin at multidecadal time scales... but such correlations are not significant in observations." Apart from sparse observations in the earlier record, this may also be a result of non-stationary teleconnections (see for example Cai et al. (2019). Pantropical climate interactions. Science, 363, eaav4236)

+ L218: "To allow a comparison between the results, also the period of variability of the historical data has been extended to 50 years...". The word 'also' is not required.

+ L223: "...but they overwhelmingly remove a linear trend...". I assume you mean here simply that a majority of the studies remove a linear trend? 'Overwhelmingly' seems to be too overwhelming a word to use. Simply stating that 'a majority remove a linear trend' is sufficient. Or 'almost all'.

+ L235: For the Fig. 5 analysis, is the Indian Ocean the only additional component for the 'Global Ocean'? In other words, if a timeseries for the Indian Ocean were added to panels 5a and 5b, would Indian+Atlantic+Pacific+Southern = Global? I'm not suggesting you add the Indian Ocean timeseries to the figure, but it might be useful to clarify this point in the text.

+ L262: "On the other hand, in the Pacific remarkable differences exist: only in the HIGH simulation OHC anomaly signals propagate equatorward around 30°N, imprinting on the global pattern." Could you please explain this further? In particular, how to see this 'imprinting'?
* * *

---

## Author Comment (AC2) · 12 Feb 2021

| **DOI:** | 10.5194/os-2020-85 |
| **Version:** | Revision |
| **Title:** | Effects of strongly eddying oceans on multidecadal climate variability in the Community Earth System Model |
| **Authors:** | André Jüling, Anna S. von der Heydt, Henk A. Dijkstra |

**Point by point reply to reviewer #2**

February 12, 2021

We thank the reviewer for their careful reading and for the useful comments on the manuscript.

**1 Reviewer Summary:**

*This study analyses the impact of model resolution on the simulation of multidecadal climate variability. 250-year simulations are run with the Community Earth System Model at high (0.1° ocean) and low (1° ocean) resolution, which are then compared to 149 years of observational data. It is found that the higher resolution run simulates larger multidecadal variability in the Atlantic and Southern Ocean (and more like observations), with little difference between the two runs in the Northern Pacific. The improvements are linked to better resolution of mesoscale ocean dynamics, and therefore larger heat content variability in the higher resolution run. Some assessment is made with regards to the impact on global mean surface temperature (GMST), but little difference is seen in multidecadal GMST variability across the two resolutions.*

*The paper is well-written, well-presented, and certainly worthy of publication in Ocean Science. The question around the impact of model resolution on the representation of multidecadal variability is likely to be of interest to the community. Clearly a lot of time has been spent on polishing the manuscript, and it is in an excellent state. There are a small number of minor points below that may require some attention.*

**2 Minor Comments:**

1. *L9: "The effect on global mean surface temperature is relatively minor". It might be better to clarify here that the effect on multidecadal GMST variability is relatively minor, since you show that there are changes to interannual variability.*
   We will make the suggested change.

2. *L34: please indicate here that Pacific Decadal Oscillation is abbreviated to PDO later (PDO is used at L58 for the first time).*
   We will add the abbreviation here.

3. *L151: appears to be the first use of 'SOM', and yet to be defined.*
   We will add the name in the introduction paragraph.

4. *L173-178: it might be useful to move the index definitions into Section 2.*
   We will move l.173-180 to the methods section, but kept Figure 2 and its description (l.180-184)

at the beginning of section 3.1.

5. *L181: "The AMV and SOM indices (in units of Kelvin) exhibit a smaller amplitude in the simulations than in the historical data". This is also true for the PDO index?*
We will rephrase the sentences describing the PDO standard deviations.

6. *L181: How much of the difference between observations and model runs can be attributed to the different data lengths? In panel d, it might be helpful to show uncertainty bars indicating the range of standard deviations for the model data, if you were to compute it in 149-year moving windows (i.e. same length as observed data).*
We will perform a block bootstrap estimate of the standard deviation.

7. *L183: "Larger PDO amplitudes ...". I don't follow this sentence. Larger PDO amplitudes with respect to what?*
(see comment 5)

8. *Fig. 2 caption: The sentence beginning with "The monthly time series of ..." requires some editing.*
(Fig. 2 caption) We will rephrase the sentence.

9. *L198: "This suggests possible correlations between the Indian and Pacific basins and the Atlantic basin at multidecadal time scales. . . but such correlations are not significant in observations." Apart from sparse observations in the earlier record, this may also be a result of non-stationary teleconnections (see for example Cai et al. (2019)*
We will elaborate on the insignificant teleconnection correlations and added a reference to Cai et al. (2019).

10. *L218: "To allow a comparison between the results, also the period of variability of the historical data has been extended to 50 years. . .". The word 'also' is not required.*
We will change the sentence to make it easier to understand.

11. *L223: "...but they overwhelmingly remove a linear trend...". I assume you mean here simply that a majority of the studies remove a linear trend? 'Overwhelmingly' seems to be too overwhelming a word to use. Simply stating that 'a majority remove a linear trend' is sufficient. Or 'almost all'.*
We will change the sentence as suggested.

12. *L235: For the Fig. 5 analysis, is the Indian Ocean the only additional component for the 'Global Ocean'? In other words, if a timeseries for the Indian Ocean were added to panels 5a and 5b, would Indian+Atlantic+Pacific+Southern = Global? I'm not suggesting you add the Indian Ocean timeseries to the figure, but it might be useful to clarify this point in the text.*
We will clarify the definitions of the ocean basins. The Global Ocean includes all oceans and marginal seas, such that it is not the sum of the Indian, Atlantic, Pacific, and Southern Oceans.

13. *L262: "On the other hand, in the Pacific remarkable differences exist: only in the HIGH simulation OHC anomaly signals propagate equatorward around $30°N$, imprinting on the global pattern." Could you please explain this further? In particular, how to see this 'imprinting'?*
We will mention the pattern that is imprinted.

**References**

Cai, Wenju et al. (2019). "Pantropical climate interactions". In: *Science* 363.6430. ISSN: 10959203. DOI: 10.1126/science.aav4236.

---

## Author Response (AR1)

| DOI:     | 10.5194/os-2020-85                                                     |
|----------|------------------------------------------------------------------------|
| Version: | Revision                                                               |
| Title:   | Effects of strongly eddying oceans on multidecadal climate variability |
|          | in the Community Earth System Model                                    |
| Authors: | André Jüling, Anna S. von der Heydt, Henk A. Dijkstra                  |

**Point by point reply to reviewer #1**

March 12, 2021

We thank the reviewer for their careful reading and for their very useful comments on the manuscript.

**1** Reviewer Summary:**

This study aimed to test the hypothesis that mesoscale resolving climate models can better simulate multidecadal climate variability of the climate system. To do so, the authors compared two simulations with constant 2000 GHG condition from the Community Earth System Model (CESM) – one with 1 degree and one with 0.1 degree ocean grid – with observational records. Specifically, the authors compared 3 climate indices between these datasets: Atlantic Multidecadal Variability (AMV), Pacific Decadal Oscillation (PDO), and Southern Ocean Mode (SOM). The authors also compared surface heat fluxes (SHF), ocean heat content (OHC), and global mean surface temperature (GMST) between these two simulations. The authors found an improvement in simulating these climate modes using the high resolution model. The low frequency of SHF and OHC were also generally larger in the high resolution model. The difference in GMST magnitude between the two simulations were timescale dependent. Based on these results, the authors asserted that using mesoscale resolved climate model could improve the representation of multidecadal climate variability in climate models.

Overall, I thought this study was very interesting and significant. In particular, this study tried to resolve issues raised regarding climate models' ability to capture 'low frequency' climate variability by a series of studies (e.g. Laepple and Huybers (2014b), Laepple and Huybers (2014a), Frankcombe et al. (2015), Cheung et al. (2017), and Parsons et al. (2017)). The approach used in this study also moved past from using idealized model to quantify the effects of small scale processes on large scale circulation. To my knowledge, the analyses done also seemed valid. Though, I believe this study would benefit from further improving the analyses done in this study and the presentation style in certain parts of the manuscript. Detail comments are below.

**2 Major Comments:**

1. It is a bit unclear what specific timescales the authors are focusing on. Throughout the article, the authors used 'multidecadal' variability without specifying the timescales. This becomes confusing when the authors discussed about specific frequencies. For instance, if high resolution model does a better job in simulating a specific frequency but not some other frequency (where both can be classified as 'multidecadal' timescale), does that mean the high resolution model is doing a better job or does it not?

(1.195) We made the time scale explicit and defined it as 10-50 years.

- 2. One major aspect of this study is the comparison of data in the frequency domain. In general, I find a little bit difficult to convince myself that the high resolution model is indeed better (or significantly different from) than the low resolution model in simulating multidecadal variability by looking at the power spectrum.
  - (a) For instance, the authors argued that the high resolution model does a better job in simulating the power spectra of all the climate modes by suggesting consistency between spectral peaks shown in observations and high resolution model. However, there are two problems.
    - First, this assumes that having the correct spectral peak is what we should aim for. However, as suggested in the text, getting the correct spectral peak that is internal to the climate system from observation is difficult. Spectral peaks can arise from incorrectly removing forced signal from observation. So, having similar spectral peaks between observation and model do not necessarily mean the model simulation is better.
    - Second, both high resolution and low resolution simulations seem to have similar spectral power. The main difference between these two simulations is related to the 'null hypothesis', where the low resolution climate model has a higher null threshold. As such, I wonder if comparing spectral peaks is a good way to quantify whether the high resolution simulation is better than the low resolution model.
  - (b) Another example is in SHF and GMST. The effects of reduced variability on ENSO band in the high resolution simulation is undoubtedly pretty clear. However, the difference between high and low resolution simulation becomes a lot less clear on longer timescales (i.e. multidecadal timescales). This again makes comparison on the frequency spectrum less powerful and makes the result less clear.

I suggest the authors can try to come up with some metrics to compare the power spectrum more quantitatively. The simplest way to compare agreements between power spectra would be to compute the spectral coherence. Alternatively, multiple studies have tried to compare the spectral slopes over a range of timescale to argue for/against of underestimation of multidecadal variability (e.g. Dee et al. (2017) and Parsons et al. (2017)). I think these examples can serve as a framework to develop more quantitative comparison between power spectrum.

We have discussed the spectral peaks and their comparison between the simulations and the observations with more nuance.

(Figs. 4, 5, 8 and ll. 193, 273, 284, 334, 351) We have also quantified the multidecadal spectral power through the spectral slope and the mean power in the multidecadal variability range (period range [10,50] years), as well as used the MV/IV ratio to quantify the relative spectral power.

3. SHF and OHC are undoubtedly related to SST. However, SHF and OHC were not mentioned explicitly throughout the introduction section and did not come up until the first paragraph of results. This introduction of SHF and OHC is rather abrupt, and it is not totally clear why a comparison of SHF and OHC between high and low resolution model is needed in the context of this study. Having more explicit description on the motivation to analyze SHF and OHC in the context of this study in the introduction part would be helpful.

(1.46) We have added an introductory paragraph about the connection of OHC to SHF and SST in the context of multidecadal variability.

4. In addition, I am also having a little bit trouble to tie the results in the SHF and OHC sections back to the SST patterns (i.e. AMV, PDO, SOM). Aside from SOM, I don't think there is enough discussion about how the more variable SHF and OHC are related to the more accurate SST patterns for AMV and PDO (or vice versa). I understand that it is outside of the scope of this study to identify the underlying mechanism that leads to an improvement in simulating multidecadal variability, but since this study analyzed SHF and OHC in tandem with these climate modes, I think it's reasonable to at least make more explicit connections between these variables in the discussion section.

(1.46) We have clarified these connections between the OHC and SHF results to the modes of multidecadal variability.

5. A large portion of results that described how the calculations were done should be moved to the method section. This includes Lines 173 – 184, Lines 249 – 256, Lines 270 – 271. There should also be discussion on how GMST is defined here.

(l.166ff.) We have moved l173-180 to the methods section, but keep Figure 2 and its description (l.180-184) at the beginning of results section 3.1.

(l.203ff.) We have moved the Ocean Heat Content definitions (originally l.249-256 & 270-271) to the methods section.

6. I suggest adding a sentence or two that describes what the implications are based on the results obtained from this study. That way, the readers can understand the major takeaways of this study from the abstract.

(1.9) We have added a sentence in the abstract and two in the discussion to make this point more explicit.

**3** Minor Comments:**

- Line 23, Zhang et al. (2019a) is probably not the best reference for this sentence. There are many other papers that are more relevant and explicit in discussing about the importance to disentangle modes of internal variability from forced changes for detection and attribution studies. Examples include: Hegerl and Zwiers (2011), Bindoff et al. (2013), and Deser et al. (2020). (1.24) We have changed the references as suggested.
- Lines 26 27, I believe there are other relevant studies that the author should cite, examples include: temperature extremes (e.g. Ruprich-Robert et al. (2018), droughts (e.g. McCabe and Palecki (2006) and Delworth et al. (2015)), hurricanes (e.g. Zhang and Delworth (2006)). For Atlantic, I think Zhang et al. (2019b) provides a good overview of societally relevant impacts induced by AMV.

(1.28) We have added the suggested references.

- Lines 31 32, I'm not sure if Atlantic Multidecadal Oscillation was named in Kushnir (1994). To my knowledge, it is more explicitly mentioned and defined in Kerr (2000), although many studies in the 1990s have already identified multidecadal variations in the North Atlantic.
   (1.35) We have adjusted the text to include both references.
- 4. Lines 122 123, it is unclear how the data is de-seasonalized. My understanding of how deseasonalize is commonly done is by first calculating the monthly climatology (Jan – Dec) and then subtracting it from the monthly data (e.g. October 2020 SST – averaged October SST). I think the first part of the sentence described this process, but the 'i.e. ...' describes something else – annual SST of each year is subtracted from monthly SST of that year. It would be great if this could be clarified.

(1.139) We have clarified that the canonical deseasonalization procedure was performed.

 Figure 2d. It should be noted that the standard deviation comparison between observation and models isn't quite 'apples' to 'apples' because their temporal lengths are different. Even though I don't expect the results would change significantly, I would try to bootstrap the results to account for sampling uncertainty.
 (210, caption Fig. 2) We performed a stationary bootstrap estimate of the standard deviation

(219, caption Fig. 2) We performed a stationary bootstrap estimate of the standard deviation. This estimate was indeed almost identical to the original one.

- Line 179, please specify which order of Butterworth filter was used.
   (1.172) We have specified that it is a second order Butterworth filter.
- Lines 198 199, it's worthy to point out Steinman et al. (2015) only focused on North Atlantic and North Pacific, and they did not look at the relationship between Atlantic, Pacific and Indian together.
   (1.226) We have specified this

(1.236) We have specified this.

- 8. Lines 218 220, please double check the sentence, something is missing. Right now, I have trouble understanding this sentence.
  (1.260) We have simplified the sentence to make it easier to understand.
- Lines 224 226, there are also studies that tried to reconstruct decadal variability in the Pacific. Since the authors mentioned paleoclimate studies in the Atlantic, I think it won't be complete without a discussion (or at least mentioning) on paleoclimate studies in the Pacific. Examples include: D'Arrigo et al. (2001), MacDonald (2005), Felis et al. (2010), and O'Mara et al. (2019). (1.266) We have added these references.
- 10. Figure 5a&b, the unit on the y-axis is wrong it should be [heat/time] since it's flux data.
  (Fig. 5) We have changed the y-axis label.
- 11. Lines 239 240, which frequencies of the spectral power for Atlantic and Pacific were integrated over?
  (1.286) We have clarified this (see also major comment 2).
- 12. Lines 244 245, citation is needed. There are studies that showed ocean dynamics (i.e. horizontal divergence) plays a significant role in driving OHC change (e.g. Roberts et al. (2017) and Small et al. (2020)).
  - (1.46) We have mentioned the role of OHC divergence in the new paragraph in the introduction.
- 13. Lines 245 246, I think this statement requires clarification. In your previous subsection, you showed that Southern Ocean (and it seems like the global ocean also) SHF exhibit a white noise whereas the Atlantic and Pacific exhibit a blue noise behavior in low frequencies. These results, to me, don't suggest a particularly strong multidecadal surface heat flux variability, at least relative to high frequency variability.

(Fig. 5) Thanks to the new spectral quantification, namely that the MV/IV ratio of HR-CESM is higher and the spectral slopes are redder in all basins and globally compared to LR-CESM, this statement is now supported.]

- 14. Line 249, please explain why and how an interpolation was done here. (1.209) We have added a sentence discussing this.
- 15. Figure 6, I'm curious as to why Indian Ocean is included here. Throughout the manuscript, this is the only place where data from the Indian Ocean was analyzed. Even though I think it's interesting to look at it, I'm not sure if it is very relevant to this manuscript, where the target is more on

global, Pacific, Atlantic, and the Southern Oceans. (Fig. 6) We have removed the Indian Ocean panels.

- 16. Lines 330 331, please double check the figure that is referred to. I think it's supposed to be Fig 6i instead of 5i. If so, Fig 6i represents depth and zonally integrated OHC but not SHF.
  (1.375) We have corrected the reference to Fig. 6i and adjust the sentence accordingly.
- 17. Line 375, note that Mann et al. (2020) showed that CMIP5 do not show multidecadal \*oscillations\* (they defined it as significant spectral peak against a null hypothesis) but not absence of multidecadal variability.

(1.420) We have rephrased the sentence to reflect this distinction.

 Line 377, 'multidecacal' should be multidecadal. (1.421) We have fixed this typo.

**References**

- Bindoff, Nathaniel L. et al. (2013). "Detection and Attribution of Climate Change: from Global to Regional". In: *Climate Change 2013 - The Physical Science Basis*. Ed. by Intergovernmental Panel on Climate Change. Vol. 9781107057. Cambridge: Cambridge University Press, pp. 867–952. ISBN: 9781107415324. DOI: 10.1017/CB09781107415324.022.
- Cheung, Anson H. et al. (2017). "Comparison of Low-Frequency Internal Climate Variability in CMIP5 Models and Observations". In: *Journal of Climate* 30.12, pp. 4763–4776. ISSN: 0894-8755. DOI: 10. 1175/JCLI-D-16-0712.1.
- D'Arrigo, R., R. Villalba, and G. Wiles (2001). "Tree-ring estimates of Pacific decadal climate variability". In: *Climate Dynamics* 18.3-4, pp. 219–224. ISSN: 0930-7575. DOI: 10.1007/s003820100177.
- Dee, S.G. et al. (2017). "Improved spectral comparisons of paleoclimate models and observations via proxy system modeling: Implications for multi-decadal variability". In: *Earth and Planetary Science Letters* 476, pp. 34–46. ISSN: 0012821X. DOI: 10.1016/j.epsl.2017.07.036.
- Delworth, Thomas L. et al. (2015). "A Link between the Hiatus in Global Warming and North American Drought". In: Journal of Climate 28.9, pp. 3834–3845. ISSN: 0894-8755. DOI: 10.1175/JCLI-D-14-00616.1.
- Deser, C. et al. (2020). "Insights from Earth system model initial-condition large ensembles and future prospects". In: *Nature Climate Change* 10.4, pp. 277–286. ISSN: 1758-678X. DOI: 10.1038/s41558-020-0731-2.
- Felis, Thomas et al. (2010). "Pacific Decadal Oscillation documented in a coral record of North Pacific winter temperature since 1873". In: *Geophysical Research Letters* 37.14, n/a–n/a. ISSN: 00948276. DOI: 10.1029/2010GL043572.
- Frankcombe, Leela M. et al. (2015). "Separating internal variability from the externally forced climate response". In: *Journal of Climate* 28.20, pp. 8184–8202. ISSN: 08948755. DOI: 10.1175/JCLI-D-15-0069.1.
- Hegerl, Gabriele and Francis Zwiers (2011). "Use of models in detection and attribution of climate change". In: *Wiley Interdisciplinary Reviews: Climate Change* 2.4, pp. 570–591. ISSN: 17577780. DOI: 10.1002/wcc.121.
- Kerr, Richard A. (2000). "A North Atlantic Climate Pacemaker for the Centuries". In: *Science* 288.5473, pp. 1984–1985.

- Kushnir, Yochanan (1994). "Interdecadal Variations in North Atlantic Sea Surface Temperature and Associated Atmospheric Conditions". In: *Journal of Climate* 7.1, pp. 141–157. ISSN: 0894-8755. DOI: 10.1175/1520-0442(1994)007<0141: IVINAS>2.0.C0; 2. arXiv: arXiv:1011.1669v3. URL: http://journals.ametsoc.org/doi/10.1175/1520-0442(1994)007{\%}3C0141: IVINAS{\%} }3E2.0.C0; 2.
- Laepple, T. and P. Huybers (2014a). "Global and regional variability in marine surface temperatures". In: *Geophysical Research Letters* 41.7, pp. 2528–2534. ISSN: 00948276. DOI: 10.1002/2014GL059345.
- Laepple, Thomas and Peter Huybers (2014b). "Ocean surface temperature variability: Large model-data differences at decadal and longer periods". In: *Proceedings of the National Academy of Sciences* 111.47, pp. 16682–16687. ISSN: 0027-8424. DOI: 10.1073/pnas.1412077111.
- MacDonald, Glen M. (2005). "Variations in the Pacific Decadal Oscillation over the past millennium". In: *Geophysical Research Letters* 32.8, p. L08703. ISSN: 0094-8276. DOI: 10.1029/2005GL022478.
- Mann, Michael E, Byron A Steinman, and Sonya K Miller (2020). "Absence of internal multidecadal and interdecadal oscillations in climate model simulations". In: *Nature Communications* 11.1, p. 49. ISSN: 2041-1723. DOI: 10.1038/s41467-019-13823-w.
- McCabe, Gregory J. and Michael A. Palecki (2006). "Multidecadal climate variability of global lands and oceans". In: *International Journal of Climatology* 26.7, pp. 849–865. ISSN: 0899-8418. DOI: 10. 1002/joc.1289.
- O'Mara, Nicholas A. et al. (2019). "Subtropical Pacific Ocean Temperature Fluctuations in the Common Era: Multidecadal Variability and Its Relationship With Southwestern North American Megadroughts". In: *Geophysical Research Letters* 46.24, pp. 14662–14673. ISSN: 0094-8276. DOI: 10.1029/2019GL084828.
- Parsons, Luke A. et al. (2017). "Temperature and Precipitation Variance in CMIP5 Simulations and Paleoclimate Records of the Last Millennium". In: *Journal of Climate* 30.22, pp. 8885–8912. ISSN: 0894-8755. DOI: 10.1175/JCLI-D-16-0863.1.
- Roberts, C. D. et al. (2017). "Surface flux and ocean heat transport convergence contributions to seasonal and interannual variations of ocean heat content". In: *Journal of Geophysical Research: Oceans* 122.1, pp. 726–744. ISSN: 21699275. DOI: 10.1002/2016JC012278. URL: http://doi.wiley.com/10.1002/2016JC012278.
- Ruprich-Robert, Yohan et al. (2018). "Impacts of the Atlantic Multidecadal Variability on North American Summer Climate and Heat Waves". In: *Journal of Climate* 31.9, pp. 3679–3700. ISSN: 0894-8755. DOI: 10.1175/JCLI-D-17-0270.1.
- Small, R. Justin et al. (2020). "What Drives Upper-Ocean Temperature Variability in Coupled Climate Models and Observations?" In: *Journal of Climate* 33.2, pp. 577–596. ISSN: 0894-8755. DOI: 10.1175/ JCLI-D-19-0295.1. URL: http://journals.ametsoc.org/doi/10.1175/JCLI-D-19-0295.1.
- Steinman, Byron A., Michael E. Mann, and Sonya K. Miller (2015). "Atlantic and Pacific multidecadal oscillations and Northern Hemisphere temperatures". In: *Science* 347.6225, pp. 988–991. ISSN: 0036-8075. DOI: 10.1126/science.1257856.
- Zhang, Liping et al. (2019a). "Natural variability of Southern Ocean convection as a driver of observed climate trends". In: *Nature Climate Change* 9.January, pp. 59–65. ISSN: 1758-6798. DOI: 10.1038/ s41558-018-0350-3. URL: https://doi.org/10.1038/s41558-018-0350-3.
- Zhang, Rong and Thomas L. Delworth (2006). "Impact of Atlantic multidecadal oscillations on India/Sahel rainfall and Atlantic hurricanes". In: *Geophysical Research Letters* 33.17, p. L17712. ISSN: 0094-8276. DOI: 10.1029/2006GL026267.
- Zhang, Rong et al. (2019b). A Review of the Role of the Atlantic Meridional Overturning Circulation in Atlantic Multidecadal Variability and Associated Climate Impacts, 2019RG000644. ISBN: 0000000284936. DOI: 10.1029/2019RG000644. URL: https://onlinelibrary.wiley.com/doi/ abs/10.1029/2019RG000644.

| DOI:     | 10.5194/os-2020-85                                                     |
|----------|------------------------------------------------------------------------|
| Version: | Revision                                                               |
| Title:   | Effects of strongly eddying oceans on multidecadal climate variability |
|          | in the Community Earth System Model                                    |
| Authors: | André Jüling, Anna S. von der Heydt, Henk A. Dijkstra                  |

**Point by point reply to reviewer #2**

March 12, 2021

We thank the reviewer for their careful reading and for the useful comments on the manuscript.

**1 Reviewer Summary:**

This study analyses the impact of model resolution on the simulation of multidecadal climate variability. 250-year simulations are run with the Community Earth System Model at high  $(0.1^{\circ} \text{ ocean})$  and low  $(1^{\circ} \text{ ocean})$  resolution, which are then compared to 149 years of observational data. It is found that the higher resolution run simulates larger multidecadal variability in the Atlantic and Southern Ocean (and more like observations), with little difference between the two runs in the Northern Pacific. The improvements are linked to better resolution run. Some assessment is made with regards to the impact on global mean surface temperature (GMST), but little difference is seen in multidecadal GMST variability across the two resolutions.

The paper is well-written, well-presented, and certainly worthy of publication in Ocean Science. The question around the impact of model resolution on the representation of multidecadal variability is likely to be of interest to the community. Clearly a lot of time has been spent on polishing the manuscript, and it is in an excellent state. There are a small number of minor points below that may require some attention.

**2 Minor Comments:**

1. L9: "The effect on global mean surface temperature is relatively minor". It might be better to clarify here that the effect on multidecadal GMST variability is relatively minor, since you show that there are changes to interannual variability.

With have reformulated this statement in light of the new quantification of the spectra.

- L34: please indicate here that Pacific Decadal Oscillation is abbreviated to PDO later (PDO is used at L58 for the first time).
   (1.39) We have added the abbreviation here.
- 3. L151: appears to be the first use of 'SOM', and yet to be defined. (1.43) We have added the name in the introduction paragraph.
- 4. L173-178: it might be useful to move the index definitions into Section 2. (l.166ff.) We have moved l.173-180 to the methods section, but kept Figure 2 and its description

(1.180-184) at the beginning of section 3.1.

- 5. L181: "The AMV and SOM indices (in units of Kelvin) exhibit a smaller amplitude in the simulations than in the historical data". This is also true for the PDO index?
  (1.221) We have rephrased the sentences describing the PDO standard deviations.
- 6. L181: How much of the difference between observations and model runs can be attributed to the different data lengths? In panel d, it might be helpful to show uncertainty bars indicating the range of standard deviations for the model data, if you were to compute it in 149-year moving windows (i.e. same length as observed data).

(1.219) We performed a stationary bootstrap estimate of the standard deviation.

- 7. L183: "Larger PDO amplitudes ...". I don't follow this sentence. Larger PDO amplitudes with respect to what? (see comment 5)
- 8. Fig. 2 caption: The sentence beginning with "The monthly time series of ..." requires some editing. (Fig. 2 caption) We have rephrased the sentence.
- 9. L198: "This suggests possible correlations between the Indian and Pacific basins and the Atlantic basin at multidecadal time scales. . . but such correlations are not significant in observations." Apart from sparse observations in the earlier record, this may also be a result of non-stationary teleconnections (see for example Cai et al. (2019) (1.236) We have elaborated on the insignificant teleconnection correlations and added a reference to Cai et al. (2019).
- 10. L218: "To allow a comparison between the results, also the period of variability of the historical data has been extended to 50 years. ...". The word 'also' is not required.
  (1.260) We have changed the sentence to make it easier to understand.
- 11. L223: "...but they overwhelmingly remove a linear trend...". I assume you mean here simply that a majority of the studies remove a linear trend? 'Overwhelmingly' seems to be too overwhelming a word to use. Simply stating that 'a majority remove a linear trend' is sufficient. Or 'almost all'. (1.262) We have changed the sentence as suggested.
- 12. L235: For the Fig. 5 analysis, is the Indian Ocean the only additional component for the 'Global Ocean'? In other words, if a timeseries for the Indian Ocean were added to panels 5a and 5b, would Indian+Atlantic+Pacific+Southern = Global? I'm not suggesting you add the Indian Ocean timeseries to the figure, but it might be useful to clarify this point in the text. (l.281) We have clarified the definitions of the ocean basins. The Global Ocean includes all oceans and marginal seas, such that it is not the sum of the Indian, Atlantic, Pacific, and Southern Oceans.
- 13. L262: "On the other hand, in the Pacific remarkable differences exist: only in the HIGH simulation OHC anomaly signals propagate equatorward around 30° N, imprinting on the global pattern." Could you please explain this further? In particular, how to see this 'imprinting'? (1.304) We have mentioned the pattern that is imprinted.

**References**

Cai, Wenju et al. (2019). "Pantropical climate interactions". In: *Science* 363.6430. ISSN: 10959203. DOI: 10.1126/science.aav4236.

---

## Author Response (AR2)

**DOI:** 10.5194/os-2020-85
**Version:** Second Revision
**Title:** Effects of strongly eddying oceans on multidecadal climate variability
in the Community Earth System Model
**Authors:** André Jüling, Anna S. von der Heydt, Henk A. Dijkstra

**Point by point reply to reviewer #1**

May 26, 2021

We thank the reviewer for their careful reading and for their very useful comments on the manuscript.

**1 Reviewer Summary:**

*I believed the author has addressed most of the comments raised in the first round of review. Overall, I think the conclusions are clearer and more convincing. I have some additional suggestions/comments that I think would help improve the manuscript.*

**2 Minor Comments:**

1. *I suggest adding uncertainties in the mean spectral power and spectral slope estimates and/or carrying out hypothesis testing to determine if the difference seen in these mean spectral powers (and spectral slope) are statistically different.*
   (l.201) We slightly changed the mean and slope estimation method to employ $\log(f)$-weighting instead of using log-equidistant bins. We use the jackknife estimator for the uncertainty at each frequency and, assuming independence, we arrive at an estimate of the standard deviation of the spectral mean. The resulting standard deviations are visualized as error bars. For the slope estimate, we report the standard error.

2. *Line 5: remove 'eventually' from the phrase 'eventually the global mean surface temperature'.*
   (l.3) Edited as suggested.

3. *Line 9: I appreciate that you tried to be more explicit about the implications of the study in the abstract and discussion. But I'm not super satisfied with the abstract part. I guess what I'm thinking is it would be great to highlight the implications of this study (mesoscale variability is important to multidecadal variability) on studies that use CMIP class models in the abstract.*
   (l.11) We added a sentence to the abstract which now makes the point that the current model generation may systematically underestimate multidecadal variability.

4. *Lines 46-57: I appreciate the addition of this paragraph to clarify the relationship between these climate modes, SHF, OHC, and GMST. However, how this paragraph is currently written is a bit awkward. Since the majority of the introduction section is about the SST modes, I would reorganize the paragraph a bit, such that it focuses on how representation of SST patterns affects our quantification of SHF, OHC, and GMST, instead of how it is currently written – how the latter*

*three metrics are related to SST.*

(l.47) Suggestion followed, we rewrote this paragraph.

5. *Lines 92-93: I would change 'In idealized non-eddying ocean models, modes of multidecadal variability exist that depend critically on the prescribed eddy diffusivity' to 'In idealized non-eddying ocean models, the existence of modes of multidecadal variability depends critically on the prescribed eddy diffusivity'.*

   (l.95) We edited the sentence as suggested.

6. *At some point in the intro, it might be worth mentioning what is the spatial scale of mesoscale just to highlight why 1 deg models do not resolve mesoscale.*

   (l.84) We included a sentence to that effect.

7. *Line 121-123: I would also add Chang et al. (2020), where they did a longer simulation and supplemented it with extremely detailed analyses.*

   (l.124) We added this excellent reference which had not been available at the time of the original writing.

8. *Line 141: For the sake of completeness, maybe it's worth to test the sensitivity of your results with other SST products also (e.g., ERSST, COBE)?*

   (l.**??**) We now include COBE and ERSST in an appendix figure. We describe the datasets in the methods and refer to differences where appropriate in the results section.

9. *Line 208: Out of curiosity, why is the HR-CESM is interpolated to specifically 0.4deg rectangular grid?*

   This is a standard interpolation routine provided by NCAR for the high resolution POP.

10. *Line 222: '(Fig. (Fig. 2d)' should be 'Fig. 2d'.*

    (l.229) changed as suggested

11. *Line 222-223: This statement is a bit confusing – it is implying that there is a causal relationship between the previous sentence and this sentence, but in reality, the larger amplitude in observed low-pass filtered PDO compared to models \*does not\* imply there is a larger proportion of spectral power >13 years. Please clarify.*

    (l.229) Indeed there is a discrepancy between that statement and the mean MV power estimates of Fig. 4 so we removed the sentence.

12. *Line 268: The correct reference for MacDonald 2005 should be MacDonald and Case 2005. Apologies for providing the incorrect reference in the previous review.*

    (l.279) We corrected the bibliography entry. It turns out this was a result of a faulty DOI database entry which Mendeley automatically retrieves.